# Developmental patterning function of GNOM ARF-GEF mediated from the cell periphery

**Maciek Adamowski[1,2], Ivana Matijević[1], Jiří Friml[1]***

[1]Institute of Science and Technology Austria, Klosterneuburg, Austria; [2]Plant Breeding and Acclimatization Institute – National Research Institute, Błonie, Poland

**Abstract** The GNOM (GN) Guanine nucleotide Exchange Factor for ARF small GTPases (ARF-GEF) is among the best studied trafficking regulators in plants, playing crucial and unique developmental roles in patterning and polarity. The current models place GN at the Golgi apparatus (GA), where it mediates secretion/recycling, and at the plasma membrane (PM) presumably contributing to clathrin-mediated endocytosis (CME). The mechanistic basis of the developmental function of GN, distinct from the other ARF-GEFs including its closest homologue GNOM-LIKE1 (GNL1), remains elusive. Insights from this study largely extend the current notions of GN function. We show that GN, but not GNL1, localizes to the cell periphery at long-lived structures distinct from clathrin-coated pits, while CME and secretion proceed normally in *gn* knockouts. The functional GN mutant variant GN$^{fewerroots}$, absent from the GA, suggests that the cell periphery is the major site of GN action responsible for its developmental function. Following inhibition by Brefeldin A, GN, but not GNL1, relocates to the PM likely on exocytic vesicles, suggesting selective molecular associations en route to the cell periphery. A study of GN-GNL1 chimeric ARF-GEFs indicates that all GN domains contribute to the specific GN function in a partially redundant manner. Together, this study offers significant steps toward the elucidation of the mechanism underlying unique cellular and development functions of GNOM.

*For correspondence:
jiri.friml@ist.ac.at

**Competing interest:** The authors declare that no competing interests exist.

## Editor's evaluation

Many plant developmental processes are mediated by gradients of the hormone auxin. GNOM is a vesicle trafficking component (an ARF-GEF) that is presumably involved in the polar transport of PINs, auxin transporters that help to establish auxin gradients. Here, the authors challenge the model that GNOM acts at intracellular compartments by providing convincing microscopy evidence that GNOM primarily acts at the cell cortex and/or the plasma membrane and they hypothesize that GNOM might establish/maintain some other unknown polarity cue at the plasma membrane. These valuable findings open new questions in the fields of plant developmental biology and plant cell biology.

## Introduction

ARF small GTPases function in eukaryotic endomembrane systems by recruiting to intracellular membranes various effectors necessary for trafficking processes, especially for vesicle formation (reviewed in *Donaldson and Jackson, 2011*; *Jackson and Bouvet, 2014*; *Singh and Jürgens, 2018*; *Yorimitsu et al., 2014*). ARFs act as GTP hydrolysis-dependent molecular switches, and their cycles of activation and deactivation are controlled by ARF regulators, GEFs (Guanine nucleotide Exchange Factors) and GAPs (GTPase Activating Proteins), respectively. In *Arabidopsis thaliana*, the ARF small

GTPase machinery functions in fundamental secretory activity at the level of Golgi apparatus (GA) and the *trans*-Golgi Network (TGN). The regulation of GA function requires the activity of the ARF-GEF GNOM-LIKE1 (GNL1) assisted by EMB30/GNOM (GN) (*Richter et al., 2007*; *Teh and Moore, 2007*), and of the ARF-GAPs AGD6-10 (ARF-GAP DOMAIN6-10) (*Min et al., 2007*; *Min et al., 2013*). The exocytic and vacuolar traffic originated from the TGN employs BIG class ARF-GEFs (*Tanaka et al., 2009*; *Richter et al., 2014*; *Xue et al., 2019*) and the ARF-GAP AGD5 (*Stefano et al., 2010*; *Sauer et al., 2013*). Notably, besides acting in these basic trafficking pathways, the ARF machinery has been ascribed a prominent function in patterned development controlled by the plant hormone auxin (*Weijers et al., 2018*). This function of the ARF machinery is, to some degree, genetically separable from the basic secretory function, and is mostly associated with the activity of the ARF-GEF GN (reviewed in *Richter et al., 2010*) and the ARF-GAP VAN3 (*Koizumi et al., 2005*; *Koizumi et al., 2000*; *Deyholos et al., 2000*; *Sieburth et al., 2006*; *Steynen and Schultz, 2003*; *Naramoto et al., 2009*; *Naramoto et al., 2010*; *Naramoto and Kyozuka, 2018*; *Adamowski and Friml, 2021a*).

The *gn/emb30* mutants were isolated in forward genetic screens aimed at the identification of early pattern formation mutants in *A. thaliana* (*Meinke, 1985*; *Mayer et al., 1991*; *Mayer et al., 1993*; *Shevell et al., 1994*). Further studies with the use of moderate and weak *gn* alleles revealed that *GN* acts also at later stages of plant development (*Geldner et al., 2004*; *Okumura et al., 2013*). *gn* mutant phenotypes include the development of fused cotyledons during embryogenesis, failure of root development at the basal embryonic pole, disorganized vascular tissue formation, defects in organization of lateral root primordia, and deficiencies in tropisms (*Mayer et al., 1991*; *Mayer et al., 1993*; *Shevell et al., 1994*; *Geldner et al., 2003*; *Geldner et al., 2004*; *Okumura et al., 2013*; *Koizumi et al., 2000*; *Miyazawa et al., 2009*; *Moriwaki et al., 2014*; *Verna et al., 2019*). Polarities of PIN1 auxin efflux carriers (reviewed in *Adamowski and Friml, 2015*) are not coordinated between cells in *gn* embryos (*Steinmann et al., 1999*). An often cited mechanism of GN action proposes that GN is required for the exocytic delivery of PIN1 to the polar (esp. basal) domain at the plasma membrane (PM) as part of PIN1's constitutive recycling between the PM and an endosome (*Geldner et al., 2001*; *Geldner et al., 2003*; *Kleine-Vehn et al., 2008*) and for PIN polarity switches during responses such as gravitropism (*Kleine-Vehn et al., 2010*; *Rakusová et al., 2011*). Beside PINs, the function of GN in auxin-mediated development encompasses other molecular targets, as well. The PM recruitment of D6 PROTEIN KINASE (D6PK), a protein kinase activating auxin efflux activity of PINs at the basal domain of the PM (*Willige et al., 2013*), is dependent on GN (*Barbosa et al., 2014*). In the context of vascular tissue patterning, GN has been suggested to influence nuclear auxin signaling, as indicated by phenotypic similarities between *gn* mutants and plants concomitantly deficient in auxin signaling and polar auxin transport (*Verna et al., 2019*); a further pathway of GN's regulation of vascular patterning relies on the regulation of plasmodesmata aperture (*Linh and Scarpella, 2022*). Finally, study of *gn* knockouts (*Shevell et al., 2000*) as well as recent forward genetic screening (*Wachsman et al., 2020*) identifies a function of GN in the regulation of cell wall composition, which may be a part of the mechanism by which GN controls developmental patterning, as well.

Altogether, studies of GN demonstrate its major function in embryonic and post-embryonic plant body development. By multiple and only partially understood mechanisms, tightly linked with the control of cellular and tissue-scale polarity, GN's activity leads to the establishment of organized polar auxin transport streams instructing patterning events during development. This activity represents a unique modification of the conserved cellular ARF machinery acting in rather housekeeping trafficking processes. To perform this function, GN acquired novel and so far elusive molecular features, absent in its GBF1-type ARF-GEF homologue GNL1 (*Richter et al., 2007*). Better understanding of the molecular and cellular bases of the unique GN function would be an important step in the unravelling of molecular mechanisms of plant developmental patterning as a whole, and would illustrate how the endomembrane system became modified in the course of evolution to provide for the requirements of a complex plant body and its adaptive life strategy.

Here, we characterize in detail the function of GN considering its molecular nature as an ARF small GTPase regulator in the cellular endomembrane system. We investigate the subcellular site of action of GN required for its role in developmental patterning, as well as GN's action in endo- and exocytic processes. We utilize direct comparisons with GNL1 to obtain precise information about the molecular function specific to GN. The internal determinants of GN function within the ARF-GEF protein itself are studied through the use of GN-GNL1 chimeric ARF-GEFs. Our findings constitute a significant

step in the elucidation of the molecular mechanism underlying the unique GN function in developmental patterning.

## Results

## GN, but not GNL1, localizes to the cell periphery at structures of unknown nature

Within the cell, the ARF-GEF GN has been originally placed at a recycling endosome (RE; *Geldner et al., 2003*), based on its co-localization with endosomal markers and with the FM4-64 fluorescent endocytic tracer dye (*Jelínková et al., 2010*) at the core of the so-called 'BFA body', an endomembrane compartment aggregation formed in the root cells of *A. thaliana* by chemical inhibition of ARF-GEFs with Brefeldin A (*Robinson et al., 2008*; *Robineau et al., 2000*). Further studies in chemically undisturbed conditions, with the use of high-resolution imaging techniques, indicated that GN localises to the GA together with its homologue GNL1, a site of action typical for GBF1-type ARF-GEFs in non-plant systems (*Naramoto et al., 2014*). In addition to the localization to the GA, GN is variably found at the PM, where it was reported to colocalize with the endocytic vesicle coat protein clathrin (*Naramoto et al., 2010*). Taken together, GN is described as localized to the GA and to the PM, presumably participating in clathrin-mediated endocytosis (CME) at the latter.

We first analysed in more detail the cellular sites of action of GN, and compared it with the localization of GNL1. For this purpose we generated new fluorescent reporter lines for GN and GNL1, expressed under *GN* promoter (*GN_{pro}:GN-GFP* and *GN_{pro}:GNL1-GFP*). We verified the functionality of the GN-expressing construct by the complementation of a *gn* null mutant allele *gn^{SALK_103014}* (*gn^s* in the following; *Figure 1—figure supplement 1A and B, E*). As a sensitive readout of GN function, we analysed root hair positioning and identified no shift toward the apical cell side, characteristic for partial *gn* loss of function (*Fischer et al., 2006*), in the complemented mutant (*Figure 1—figure supplement 1D*). We also confirmed the observation that expression of additional copies of *GNL1* cannot complement the *gn* mutant (*Figure 1—figure supplement 1C*; *Richter et al., 2007*), verifying that GNL1 does not possess a function in developmental patterning shared with GN, an important assumption of the present work.

Having confirmed this, we compared the subcellular localizations of *GN_{pro}:GN-GFP* and *GN_{pro}:GNL1-GFP,* expressed in the wild-type background, by confocal laser scanning microscopy (CLSM) in the epidermis of seedling root apical meristems (RAMs). In two independent transgenic lines expressing each construct, with settings aimed at the detection of weak signals, GN-GFP was observed at the PM in all seedlings analysed, while GNL1-GFP was never found localizing to the PM (*Figure 1A* and *Figure 1—figure supplement 2A*). Beside this difference, a larger portion of GN was observed in the cytosol relative to punctate signals representing GA, in contrast to a strong binding of GNL1 to these organelles. In comparison with previously reported observations (*Naramoto et al., 2010*), our new *GN_{pro}:GN-GFP* lines suggest that the localization of GN to the PM is a typical, rather than a rare, phenomenon.

Next, we compared the localizations of GN-GFP and GNL1-GFP using Total Internal Reflection Fluorescence (TIRF) microscopy in the early elongation zone of seedling roots (*Figure 1B and C*, *Figure 1—figure supplement 2B*, *Figure 1—videos 1 and 2*). This method is used for the observation of fluorescent reporters localized at the PM and in the cytosol directly underneath, at high magnification. In two independent transgenic lines expressing each construct, we observed the localization of both ARF-GEFs to the GA and often to weakly fluorescent, small and dynamic structures, probably residing in the cytosol (*Figure 1B*, *Figure 1—videos 1 and 2*).

In turn, only GN, but not GNL1, was found at PM-localized structures characterized by a relatively bright fluorescence and often by high stability over time (*Figure 1—video 1*, *Figure 1B and C*, *Figure 1—figure supplement 2B*). In 100 s-long time lapses, many of the structures could be seen docked in the PMs throughout the whole time course of observation, indicating that residence times often exceeded 100 s. The structures exhibited limited lateral diffusion in plane of the PM (*Figure 1C*, *Figure 1—figure supplement 4A*). These structures represent a subcellular localization site specific for GN, but not for GNL1, and as such, are a potential site of action connected with the exclusive GN's function in developmental patterning. While the structures are likely bound to the PM, the TIRF

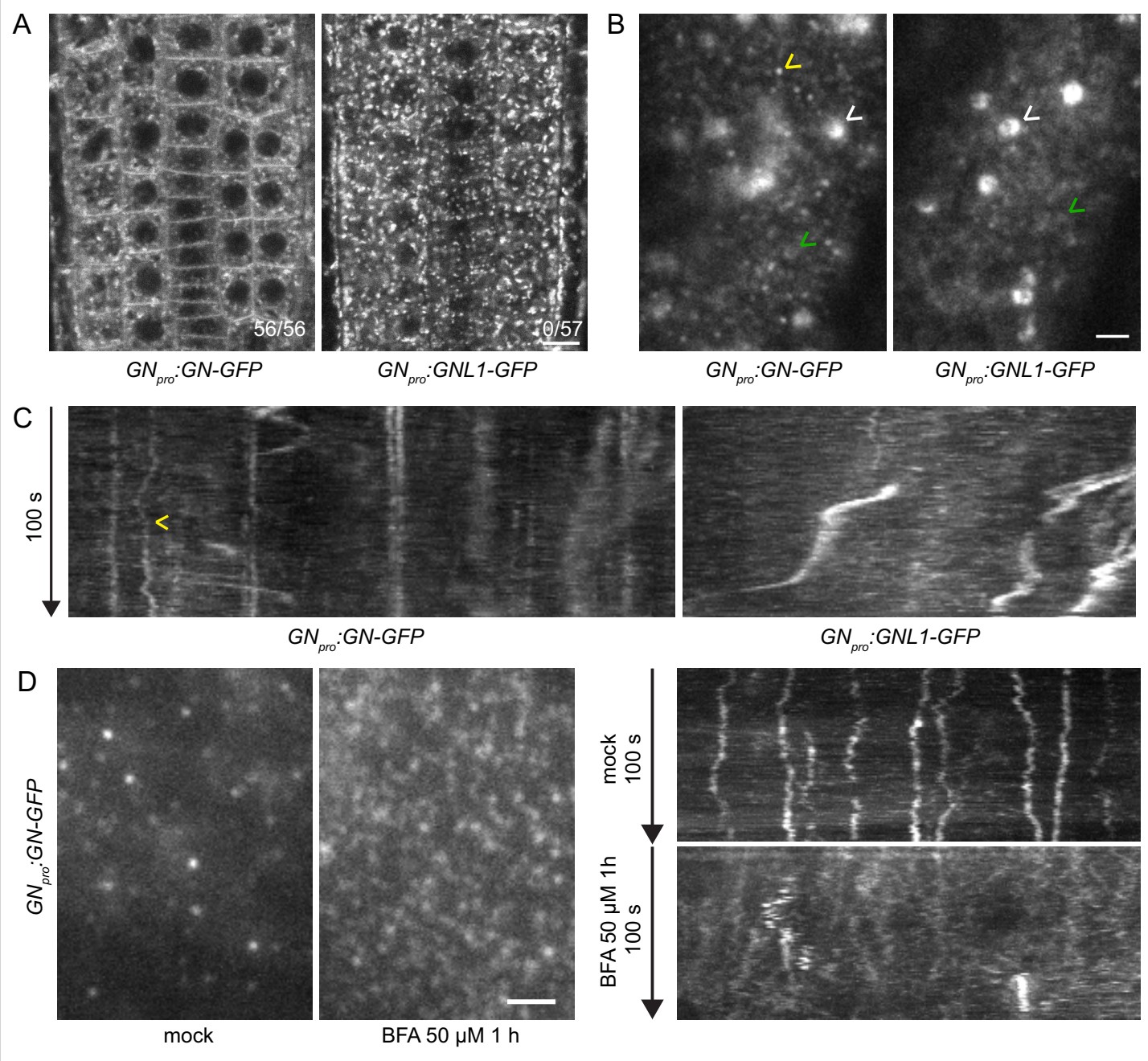

**Figure 1.** GN localizes to the cell periphery at often stable punctate structures. (**A**) CLSM images of GN-GFP and GNL1-GFP in the epidermis of seedling RAMs. In contrast with GNL1-GFP, GN-GFP localizes to the PM. Numbers in bottom right indicate ratios of RAMs with PM signals. Images are representative of data from two independent transgenic lines. Scale bar – 10 μm. (**B**) TIRF images of GN-GFP and GNL1-GFP in the epidermis of early elongation zone of seedling roots. In contrast with GNL1-GFP, the PM signal of GN-GFP consists of relatively bright punctate structures (yellow arrowhead). Both GN-GFP and GNL1-GFP are seen at the GA (white arrowheads) and are present as dynamic, weakly labeled entities likely located in the cytoplasm (green arrowheads). Scale bar – 2 μm. (**C**) Kymograph maximum projections of 15 μm sections from TIRF movies of GN-GFP and GNL1-GFP, showing GN-GFP-positive punctate structures (arrowhead). The structures are typically characterized by high stability and tend to exhibit limited lateral movement in the plane of the PM. Presence of signals exhibiting any stability is rare in GNL1-GFP. Thick, irregular shapes in both movies are traces of GA moving in the cytoplasm. (**D**) TIRF images and kymograph maximum projections of 15 μm sections of TIRF movies of GN-GFP expressed in *GN_pro_:GN-GFP VAN3_pro_:VAN3-mRFP,* captured in etiolated hypocotyl epidermis following a treatment with BFA at 50 μM for 1 hr. After a treatment with BFA, the stable structures are mostly lost, and punctate signals of high density and typically brief lifetimes are observed. Scale bar - 2 μm.

The online version of this article includes the following video and figure supplement(s) for figure 1:

**Figure supplement 1.** *gn^s* mutant complementation data.

*Figure 1 continued on next page*

*Figure 1 continued*

**Figure supplement 2.** GN-GFP and GNL1-GFP subcellular localization data from two transgenic lines analyzed.

**Figure supplement 3.** TIRF colocalization of GN-GFP and VAN3-mRFP following a BFA treatment.

**Figure supplement 4.** Quantitative data on GN-positive peripheral structures.

**Figure 1—video 1.** TIRF time lapse of GN-GFP in seedling root epidermis.
https://elifesciences.org/articles/68993/figures#fig1video1

**Figure 1—video 2.** TIRF time lapse of GNL1-GFP in seedling root epidermis.
https://elifesciences.org/articles/68993/figures#fig1video2

**Figure 1—video 3.** TIRF time lapse of GN-GFP in hypocotyl epidermis following a mock treatment.
https://elifesciences.org/articles/68993/figures#fig1video3

**Figure 1—video 4.** TIRF time lapse of GN-GFP in hypocotyl epidermis following a treatment with BFA 50 µM for 1 hr.
https://elifesciences.org/articles/68993/figures#fig1video4

**Figure 1—video 5.** TIRF time lapse colocalization of GNOM-GFP and VAN3-mRFP in hypocotyl epidermis following a treatment with BFA 50 µM for 1 hr, green channel.
https://elifesciences.org/articles/68993/figures#fig1video5

**Figure 1—video 6.** TIRF time lapse colocalization of GNOM-GFP and VAN3-mRFP in hypocotyl epidermis following a treatment with BFA 50 µM for 1 hr, red channel.
https://elifesciences.org/articles/68993/figures#fig1video6

**Figure 1—video 7.** TIRF time lapse colocalization of GNOM-GFP and VAN3-mRFP in hypocotyl epidermis following a treatment with BFA 50 µM for 1 hr, merged channels.
https://elifesciences.org/articles/68993/figures#fig1video7

microscopy setup used does not allow us to exclude a cortical localization, i.e., a localization at a certain distance from the PM.

The localization of GN at the PM is strongly enhanced following its inhibition by BFA (*Naramoto et al., 2010*; Figure 5A). We employed the *GN_pro:GN-GFP VAN3_pro:VAN3-mRFP* double reporter line and observed an almost complete disappearance of the stable GN-positive structures following a treatment with BFA at 50 µM for 1 hr. Instead, BFA-inhibited GN appeared as very dense punctate signals, characterized by rapid dynamics at the PM (*Figure 1D*, *Figure 1—figure supplement 4B* and *Figure 1—videos 3 and 4*). These signals were reminiscent of the dense and dynamic PM localization of the ARF-GAP VAN3 in chemically undisturbed conditions (*Adamowski and Friml, 2021a*), but the BFA-induced GN signals did not colocalize with the similar signals of VAN3 (*Figure 1—figure supplement 3* and *Figure 1—videos 5–7*).

Taken together, our new observations of GN localization patterns, in direct comparison with its close homologue GNL1, point to the existence of structures of unknown nature present at the cell periphery, which may be the sites of action of GN in developmental patterning.

## The functional GN^fwr mutant variant localizes to the cell periphery but not to the Golgi

Given the notion that the cell periphery may be the site of action relevant for the developmental patterning function of GN, we were interested in analysing a previously described allele *fewerroots* (GN^fwr; *Okumura et al., 2013*). *fwr* is a weak *gn* mutant allele, carrying a single amino acid substitution in the HDS2 (HOMOLOGY DOWNSTREAM OF SEC7 2) domain. The mutant develops relatively normally, with the exception of a reduced ability to form lateral roots (LRs; *Okumura et al., 2013*), and a moderate phenotype in vascular network patterning (*Verna et al., 2019*). Thus, GN^fwr may be considered a relatively well-functioning GN protein variant. Interestingly, it was reported that GN^fwr localizes to the PM, instead of the GA, in undisturbed conditions (*Okumura et al., 2013*). To explore the relevance of GN's localization at the cell periphery, we compared directly the localization patterns and functionality of GN^fwr and of wild-type GN.

First, we tested the functionality of GN^fwr-GFP protein fusions by the complementation of *gn^s* mutants with a *GN_pro:GN^fwr-GFP* construct. Two independent *gn^s GN_pro:GN^fwr-GFP* lines were phenotypically indistinguishable from control *gn^s GN_pro:GN-GFP* lines, and from wild-type plants, during seedling development and at adult stages (*Figure 2A and B*, *Figure 1—figure supplement 1E*), and

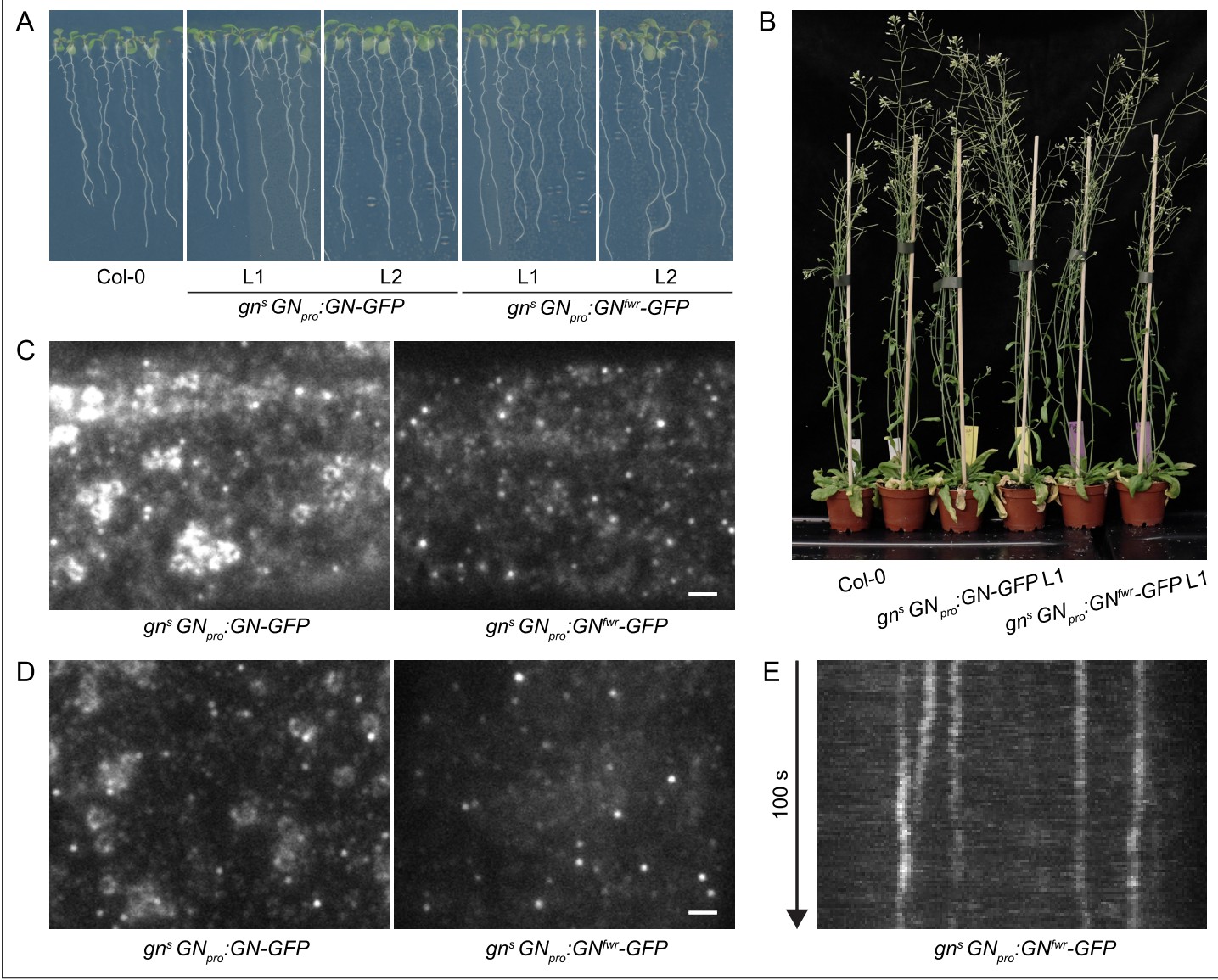

**Figure 2.** A functional GN$^{fwr}$-GFP variant localizes to punctate structures at the cell periphery, but not to the GA. (**A**) Eight-day-old seedlings of *gn$^s$* mutants complemented with *GN$_{pro}$:GN-GFP* and *GN$_{pro}$:GN$^{fwr}$-GFP* transgenes. Two independent lines of each complemented mutant are shown. (**B**) Adult *gn$^s$* mutants complemented with *GN$_{pro}$:GN-GFP* and *GN$_{pro}$:GN$^{fwr}$-GFP* transgenes. (**C**) and (**D**) TIRF images of GN-GFP and GN$^{fwr}$-GFP expressed in complemented *gn$^s$* background in hypocotyls of etiolated seedlings (**C**) and in early elongation zone of seedling roots (**D**). GN$^{fwr}$ localizes to GN-specific structures at the PM, but not to the GA. Scale bars – 2 μm. (**E**) Kymograph maximum projection of a 15 μm section from a TIRF movie of GN$^{fwr}$-GFP showing stable, PM-localized punctate structures.

The online version of this article includes the following video and figure supplement(s) for figure 2:

**Figure supplement 1.** Subcellular localization of GN-GFP and GN$^{fwr}$-GFP at basal end of young trichoblasts.

**Figure 2—video 1.** TIRF time lapse of GNfwr-GFP in gns seedling root epidermis.

https://elifesciences.org/articles/68993/figures#fig2video1

presented normal root hair outgrowth positioning at basal cell sides (*Figure 1—figure supplement 1D*). These observations confirm that GN$^{fwr}$ is a highly functional GN variant.

Next, using TIRF, we directly compared the localization patterns of GN-GFP and GN$^{fwr}$-GFP, expressed in complemented mutant backgrounds, in the epidermis of etiolated hypocotyls and of the early elongation zone of roots (*Figure 2C–E* and *Figure 2—video 1*). Like GN-GFP, GN$^{fwr}$-GFP localized to punctate structures at the cell periphery characterized by a relatively high signal intensity and high stability. The density of GN$^{fwr}$-GFP structures was slightly higher than those of GN-GFP, while

their lateral displacement was similar to wild-type GN (*Figure 1—figure supplement 4A and C*). In contrast to GN-GFP, GN^fwr^-GFP was never found localized to the GA in neither of the organs (31 time lapses and 47 single snapshots captured from 50 seedlings in the course of experiments). Considering that the observations were made in successfully complemented *gn* null mutants, the localization of GN^fwr^-GFP to the peripheral structures, but not to the GA, constitutes strong evidence that these structures are the site of GN action responsible for its function in developmental patterning. While both GN-GFP (*Figure 1A*, 5A) and GN^fwr^-GFP (Figure 5C) are also detected as diffuse cytosolic signals, the cell periphery is the only site where both specifically accumulate, supporting it as a central site of activity.

We additionally assessed localization patterns of GN-GFP and GN^fwr^-GFP during root hair initiation, to ascertain the validity of our observations in a context of a known GN function carried out in the epidermis, i.e., root hair positioning (*Fischer et al., 2006*) and correctly mediated by the used fluorescent protein fusions (*Figure 1—figure supplement 1D*). We employed TIRF imaging at basal ends of young trichoblasts, and documented localization of GN-GFP and GN^fwr^-GFP to the cell periphery-associated structures, presumably during the process of root hair positioning (*Figure 2—figure supplement 1A and B*).

The difference in phenotypes between *gn* knockout mutants complemented by GN^fwr^-GFP expression, where normal phenotypes were found by all applied criteria, and the original *fwr* mutant (*Okumura et al., 2013*), might be explained by a higher level of GN^fwr^ protein expression, or altered expression patterns, in the transgenic lines. In our view, even if, as the *fwr* phenotype indicates, GN^fwr^ is considered as an incompletely effective GN variant, unable to promote normal LR formation or fully normal vascular patterning, but functional in embryonic and most of post-embryonic development, the presented argument that cell periphery, rather than the GA, is the major site of GN action in developmental patterning, remains valid. Still, we cannot exclude the possibility that other aspects of developmental patterning, not evaluated here, may have different localization requirements.

In summary, the comparison of subcellular localizations of GN and GNL1, as well as the localization pattern of the GN mutant variant GN^fwr^, suggest that the developmental function of GN is, at least to a major degree, associated with the structures of unknown nature localized at, or in a close distance to, the PM.

## GN-positive structures are distinct from clathrin-coated pits and CME functions in the absence of GN

The localization of GN at the PM was previously associated with a proposed function of GN, and the ARF small GTPase machinery as a whole, in CME (*Naramoto et al., 2010*). Yet, the GN-positive structures do not resemble clathrin-coated pits (CCPs) forming at the PM in several characteristics: typically very long lifetimes at the PM, lateral mobility not observed with CCPs, and low density. To clarify this discrepancy, we verified the co-localization patterns of GN and clathrin at the PM by capturing TIRF time lapses of the *GN_{pro}:GN-GFP 35S_{pro}:CLC-mKO (CLATHRIN LIGHT CHAIN-monomeric Kusabira Orange)* double marker line, which marks the CLC subunit of the clathrin coat. In TIRF time lapses, the GN-positive structures were clearly distinct from CCPs labeled by CLC-mKO (*Figure 3A and B*, *Figure 3—videos 1–3*). As a comparison, we performed TIRF co-localization of fluorescent reporters for two components of CCPs, TPLATE-GFP and AP2A1-TagRFP, where co-localization was readily detected (*Figure 3—figure supplement 1*; *Gadeyne et al., 2014*). Together, the observations are not supportive of a functional link between GN function at the PM, and CME.

Next, as a test of GN requirement in CME, we crossed fluorescent reporter protein fusions for clathrin (*CLC2_{pro}:CLC2-GFP*) and for the TPLATE complex, a component of CCPs (*LAT52_{pro}:T-PLATE-GFP*; *Gadeyne et al., 2014*) into *gn^s* mutant background, and used TIRF microscopy to assess the activity of CME in the absence of GN function. We imaged the epidermis of the middle region of etiolated *gn* seedlings, corresponding to the hypocotyl, and as a control, we used etiolated hypocotyls of wild-type siblings from the segregating *gn^s* populations. In terms of the density of CCPs distributed at the PMs, labeled either by CLC2-GFP or TPLATE-GFP, the *gn^s* mutants were remarkably normal (*Figure 3C and E*, and *Figure 3—videos 4–7*). When lifetimes of the individual CLC2-GFP and TPLATE-GFP foci at the PMs were measured to evaluate the temporal characteristics of the endocytic vesicle formation events, the distribution of TPLATE-GFP foci lifetimes in *gn^s* mutants was very similar to wild-type controls (*Figure 3F*), while the distribution of CLC2-GFP foci lifetimes varied slightly,

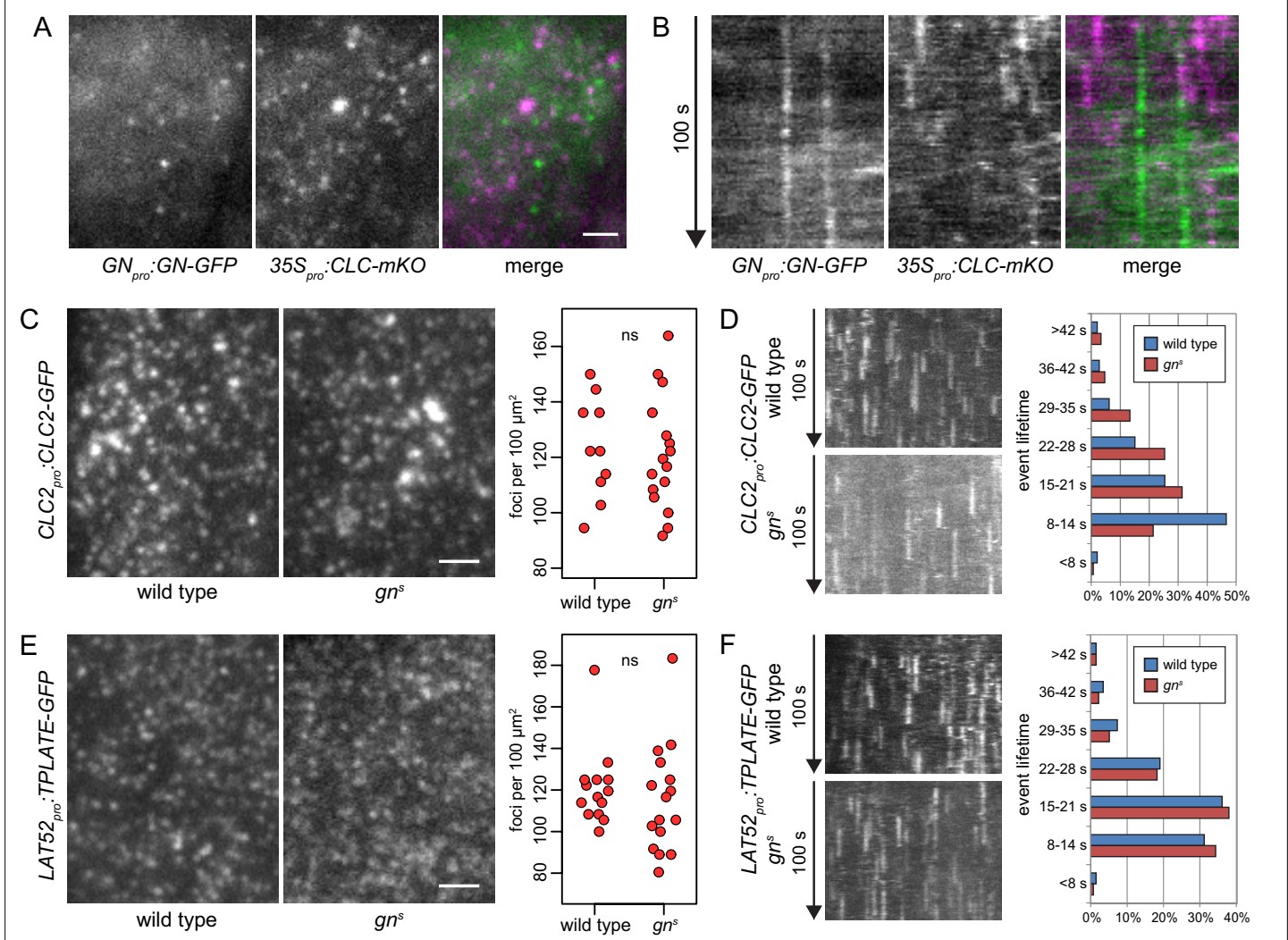

**Figure 3.** GN-positive structures do not contain clathrin, and CME is normal in *gn*[s]. TIRF images (**A**) and kymographs (**B**) of GN-GFP and CLC-mKO in the epidermis of early elongation zone of seedling roots. The GN-positive structures at the PM are distinct from CCPs. Scale bar – 2 µm. TIRF images of CLC2-GFP (**C**) and TPLATE-GFP (**E**) in middle regions of etiolated *gn*[s] seedlings and in hypocotyls of etiolated wild-type controls. Scale bars – 2 µm. Graphs show quantifications of foci densities at the PM, each data point representing a measurement from a single movie. CLC2-GFP wild type: 123±17 foci per 100 µm² (mean ± s.d.), n=10; *gn*[s]: 121±20 foci per 100 µm², n=16. TPLATE-GFP wild type: 121±18 foci per 100 µm², n=14; *gn*[s]: 115±25 foci per 100 µm², n=16. Values were compared using *t* tests, ns – not significant. Kymographs from TIRF movies representing dynamics of CLC2-GFP (**D**) and TPLATE-GFP (**F**) in middle regions of etiolated *gn*[s] seedlings and in hypocotyls of etiolated wild-type controls. Histograms show distributions of lifetimes of single endocytic events. CLC2-GFP wild type n=146, *gn*[s] n=150; TPLATE-GFP wild type n=205, *gn*[s] n=137.

The online version of this article includes the following video and figure supplement(s) for figure 3:

**Figure supplement 1.** Control TIRF co-localization of CME markers TPLATE-GFP and AP2A1-TagRFP.

**Figure 3—video 1.** TIRF time lapse colocalization of GN-GFP and CLC-mKO in seedling root epidermis, green channel.
https://elifesciences.org/articles/68993/figures#fig3video1

**Figure 3—video 2.** TIRF time lapse colocalization of GN-GFP and CLC-mKO in seedling root epidermis, red channel.
https://elifesciences.org/articles/68993/figures#fig3video2

**Figure 3—video 3.** TIRF time lapse colocalization of GN-GFP and CLC-mKO in seedling root epidermis, merged channels.
https://elifesciences.org/articles/68993/figures#fig3video3

**Figure 3—video 4.** TIRF time lapse of CLC2-GFP in wild-type hypocotyl epidermis.
https://elifesciences.org/articles/68993/figures#fig3video4

**Figure 3—video 5.** TIRF time lapse of CLC2-GFP in gns hypocotyl epidermis.
https://elifesciences.org/articles/68993/figures#fig3video5

*Figure 3 continued on next page*

*Figure 3 continued*

**Figure 3—video 6.** TIRF time lapse of TPLATE-GFP in wild-type hypocotyl epidermis.
https://elifesciences.org/articles/68993/figures#fig3video6

**Figure 3—video 7.** TIRF time lapse of TPLATE-GFP in gns hypocotyl epidermis.
https://elifesciences.org/articles/68993/figures#fig3video7

showing a higher representation of short events (8–14 s) in the wild type compared with *gn*ˢ seedlings (*Figure 3D*). Yet, we do not consider this minor difference in one of the two studied markers to be indicative of a defective CME in *gn* null mutants: Should GN function rely on controlling CME, the strong deficiency in overall growth and development in *gn* knockouts would be expected to be associated with a similarly major defect in CME.

In summary, the lack of co-localization of GN-specific structures at the cell periphery with CCPs, as well as lack of significant deficiencies in CME in *gn* knockout seedlings, argue that the function of GN at or near the PM is not directly linked to the endocytic process.

## Bulk secretion proceeds normally in the absence of GN

To complement the experiments scrutinizing the GN function in CME, we tested the function of the secretory pathway in *gn*ˢ mutants. For this purpose we crossed *gn*ˢ with *secRFP*, a line expressing a variant of red fluorescent protein (RFP) containing a signal peptide guiding it into the lumen of the endoplasmic reticulum (ER), ultimately causing the protein to be secreted to the apoplast. Using CLSM, we found that secRFP was normally secreted to the apoplast in basal ends of *gn*ˢ seedlings (*Figure 4A*), and in other parts of the seedling (*Figure 4B*), suggesting a functional secretory pathway. secRFP secretion was also normal in control 2-day-old wild-type seedlings with roots closer resembling *gn*ˢ basal ends (*Figure 4—figure supplement 1A*). Apparent intracellular fluorescent signals in *gn*ˢ basal ends represented autofluorescence, likely of plastids, as these signals were also observed in *gn*ˢ not expressing *secRFP* (*Figure 4—figure supplement 1B*).

Taken together, both CME and bulk secretion function correctly in the *gn*ˢ mutant. The function of GN in the endomembrane system, which translates into its role in auxin-mediated developmental patterning, is thus rather associated with the control of specific targets, candidates for which are PINs, or unknown regulators of PIN polarity (*Steinmann et al., 1999*; *Zhang et al., 2020*); D6PK, or

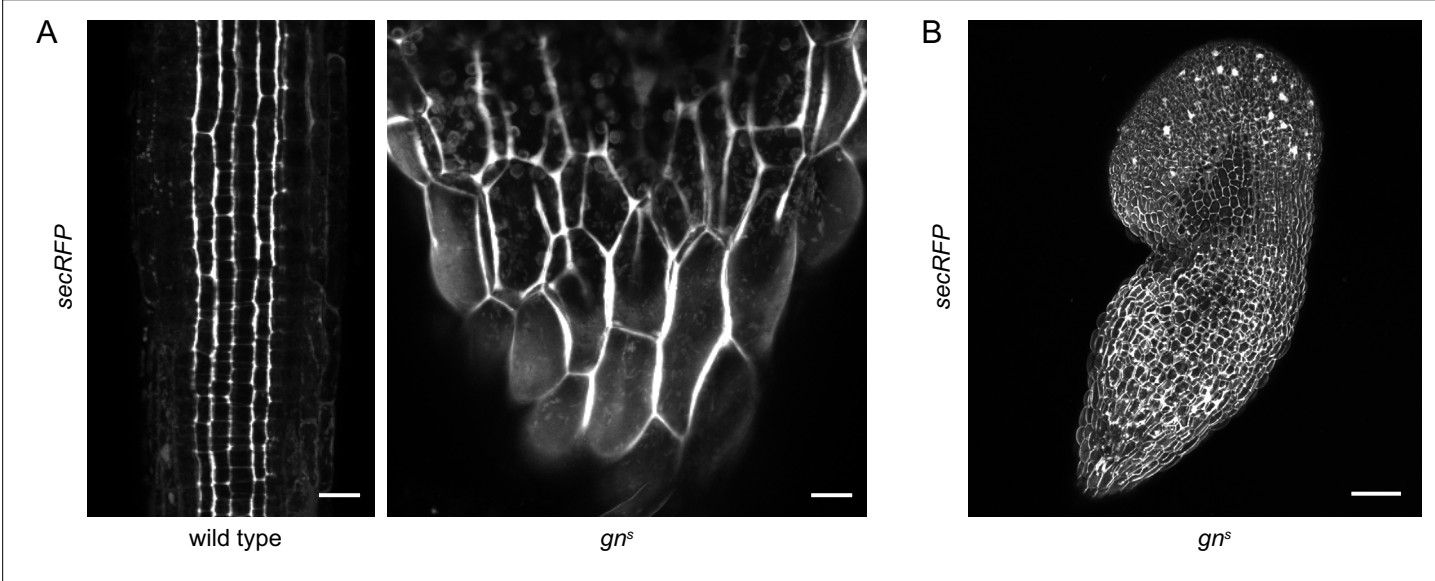

**Figure 4.** Bulk secretion is normal in *gn*ˢ. (**A**) Maximum projections of z-stacks of CLSM images of secRFP in RAMs of wild-type seedlings and in basal ends of *gn*ˢ seedlings. Scale bars - 20 μm. (**B**) Maximum projection of a z-stack of CLSM images of secRFP in a whole *gn*ˢ seedling. Scale bar - 200 μm.

The online version of this article includes the following figure supplement(s) for figure 4:

**Figure supplement 1.** Additional controls for secRFP secretion.

unknown regulators of its association with the PM (*Barbosa et al., 2014*); and pectin components of the cell wall or proteins regulating their deposition (*Shevell et al., 2000*; *Wachsman et al., 2020*). As a further exploration of GN's activity in the endomembrane system, in the following, we describe a GN-specific exocytic event, which is indicative of GN's selective association with unknown components of the secretory pathway, possibly regulators involved in GN's cellular activity that translates into developmental patterning.

### BFA-induced exocytic relocation of GN, but not GNL1, indicates GN-specific molecular interactions

It has been previously observed that GN association with the PM is increased following its inhibition with BFA (*Naramoto et al., 2010*). We re-assessed this artificially induced phenomenon and gained additional insights into the specificity and the likely site of GN action in the endomembrane system when compared with GNL1. We re-evaluated the observation using the previously generated

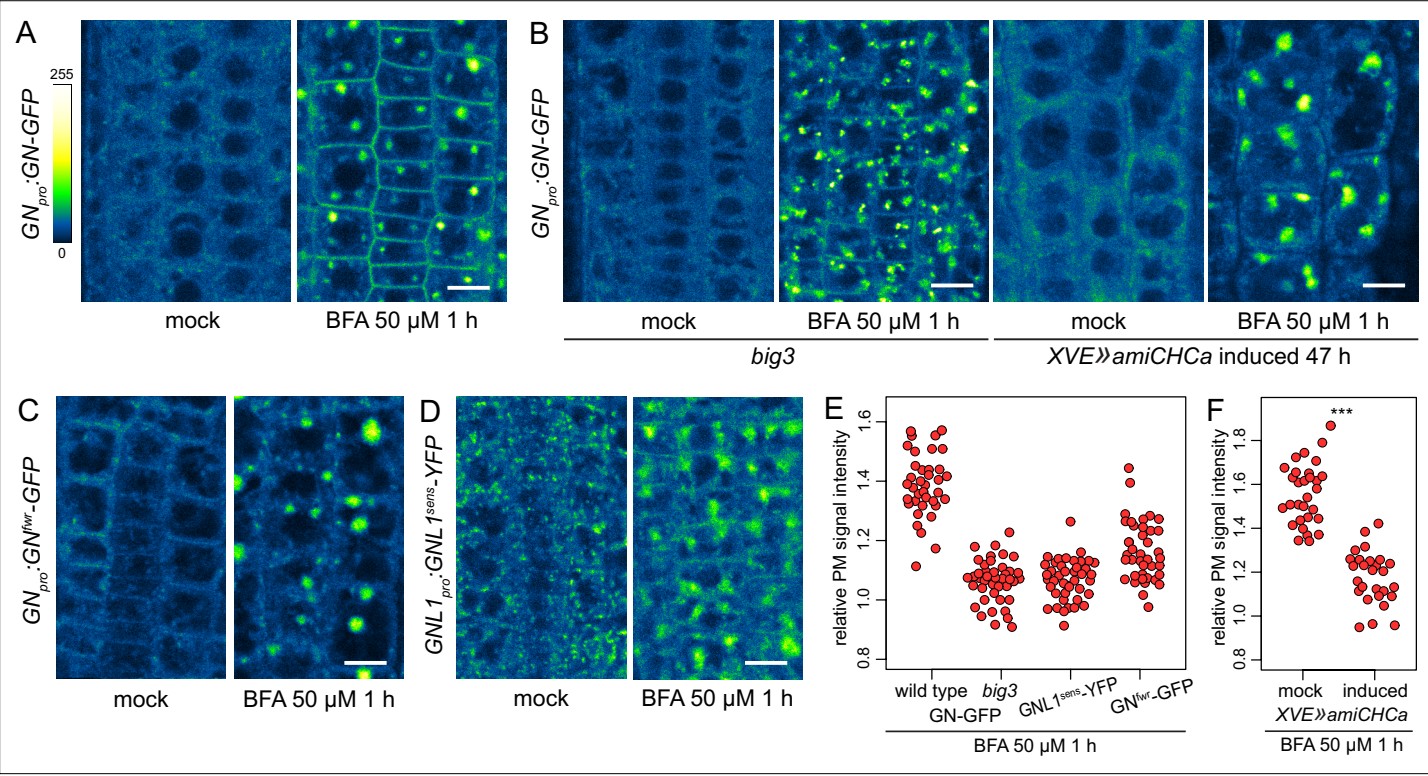

**Figure 5.** A BFA-induced, GN-specific exocytic event. (**A**) CLSM images of GN-GFP in seedling RAM epidermis following a treatment with BFA at 50 μM for 1 hr. After a treatment with BFA, GN-GFP locates to the core of BFA bodies and strongly localizes to the PM. Scale bar - 10 μm. (**B**) CLSM images of GN-GFP in seedling RAM epidermis of *big3* and of induced *XVE»amiCHCa*, following treatments with BFA at 50 μM for 1 hr. The relocation of GN-GFP to the PM is abolished to a large degree in *big3* and completely in *XVE»amiCHCa*. Scale bars - 10 μm. Treatments of non-induced *XVE»amiCHCa* lines are shown in *Figure 5—figure supplement 1B*. (**C**) CLSM images of GN^fwr^-GFP in seedling RAM epidermis following a treatment with BFA at 50 μM for 1 hr. After a treatment with BFA, GN^fwr^-GFP binds to the core of BFA bodies but does not localize to the PM. Scale bar - 10 μm. (**D**) CLSM images of a BFA-sensitive variant of GNL1-YFP in seedling RAM epidermis following a treatment with BFA at 50 μM for 1 hr. The images show GNL1^sens^-YFP expressed in a wild-type background in *gnl1 GNL1_pro_:GNL1^sens^-YFP* x Col-0 F1. After a treatment with BFA, GNL1^sens^-YFP locates to the core of BFA bodies and weakly to the PM. Scale bar - 10 μm. (**E**) Graph showing quantifications of PM signal intensities relative to total RAM signals of GN-GFP in wild type and *big3* mutants, of GNL1^sens^-YFP, and of GN^fwr^-GFP, following treatments with BFA at 50 μM for 1 h. GN-GFP wild type: 1.39±0.11 (mean ± s.d.), n=37; GN-GFP *big3*: 1.06±0.07, n=42; GNL1^sens^-YFP: 1.07±0.07, n=42; GN^fwr^-GFP: 1.16±0.10, n=40. The graph has an indicative purpose only, as comparison of quantified ratios does not reflect the visual observations precisely. This is due to differences in the samples such as the expression of different marker proteins, or alterations in BFA body structures in *big3*. (**F**) Graph showing quantifications of PM signal intensities relative to total RAM signals of GN-GFP following treatments with BFA at 50 μM for 1 h in control and induced *XVE»amiCHCa* lines. Mock: 1.56±0.07 (mean ± s.d.), n=30; induced: 1.18±0.12, n=27. Values were compared using a *t* test, p<0.0001.

The online version of this article includes the following figure supplement(s) for figure 5:

**Figure supplement 1.** A BFA-induced, GN-specific exocytic event.

*GN~pro~:GN-GFP VAN3~pro~:VAN3-mRFP* reporter line. We note that this GN-GFP fluorescent reporter appears to localize to the PM only variably when observed with CLSM in undisturbed conditions (*Naramoto et al., 2010* and our observations), in contrast to the GN-GFP reporters generated in the present study, where a localization to the PM is more consistent (*Figure 1A*). That said, the previously published GN-GFP marker does appear in TIRF microscopy in the form of typically stable structures (*Figure 1D* and *Figure 3A and B*) just like the newly generated GN-GFP and GN^fwr^-GFP markers (*Figure 1C* and *Figure 2E*). It is at present unclear why the PM detection of GN-GFP with CLSM is more variable. We note that the detection of CLC markers by CLSM at the PMs of RAM epidermis is similarly limited (e.g. *Adamowski et al., 2018*, *Adamowski et al., 2021b*) despite CME being a housekeeping process.

In BFA-treated conditions, besides localizing to the PM (*Figure 5A*), GN-GFP was strongly associated with cell plates in dividing cells (*Figure 5—figure supplement 1A*). The presence at cell plates indicated that the localization pattern of GN following BFA treatments may represent the outcome of an anterograde secretory movement of GN through the endomembrane system: this pathway leads to the PM in interphase cells, but preferentially to the cell plate during cell division (*Richter et al., 2014*). Such anterograde traffic of BFA-inhibited GN would be consistent with the previous observation that BFA-inhibited GN relocates from the GA to the TGN compartments (*Naramoto et al., 2014*). This can be perceived as a first step in an anterograde translocation of BFA-inhibited GN from the GA, through the TGN, and ultimately to the PM or the cell plate by exocytosis. As a cytosolic protein peripherally binding with membranes, we presume GN to translocate through the secretory pathway not like a cargo, but rather continuously externally bound to membranes of organelles and secretory vesicles in an abnormal fashion, likely due to its chemically inhibited state.

To test the notion that the localization of the BFA-inhibited GN at the PM represents an outcome of an exocytic event, we crossed the *GN~pro~:GN-GFP VAN3~pro~:VAN3-mRFP* line with *big3*, a mutant lacking the only BFA-resistant homologue within the BIG ARF-GEF family required for exocytosis and cell plate-directed transport from the TGN (*Richter et al., 2014*; *Xue et al., 2019*). A BFA treatment in this line causes a block of exocytosis from the TGN due to the inhibition of the remaining, BFA-sensitive ARF-GEFs of the BIG class (*Richter et al., 2014*). If the relocation of GN to the PM occurs through exocytosis, it is expected to be inhibited in this mutant background. Indeed, following a treatment with BFA at 50 µM for 1 hr in *big3*, the relocation of GN-GFP to the PM was minimal (*Figure 5B and E*). As an independent verification, we introduced into *GN~pro~:GN-GFP VAN3~pro~:VAN3-mRFP* an estradiol-inducible artificial microRNA construct down-regulating the expression of *CLATHRIN HEAVY CHAIN (CHC)* genes, which encode a key structural component of the clathrin coats. Following silencing of *CHC*, secretion is inhibited and secretory cargoes are re-routed to the vacuole (*Adamowski et al., 2021b*). In *XVE»amiCHCa*, the BFA-induced relocation of GN-GFP to the PM too, was inhibited (*Figure 5B and F*, and *Figure 5—figure supplement 1B*). This finding further supports our conclusion that BFA-inhibited GN undergoes relocation to the PM on exocytic vesicles.

To address the specificity of the induced exocytosis of GN, we tested whether GNL1 undergoes a similar BFA-induced relocation. GNL1 is resistant to BFA due to a natural variation in the sequence of its catalytic SEC7 domain (*Richter et al., 2007*). As expected, following a BFA treatment, the native, BFA-resistant GNL1-GFP was retained at the GA, being distributed at the periphery of BFA bodies (*Figure 5—figure supplement 1C*). For a meaningful comparison with GN, we analysed the reactions of the engineered BFA-sensitive GNL1 variant, GNL1^sens^-YFP (*Richter et al., 2007*). When assayed in *gnl1 GNL1~pro~:GNL1^sens^-YFP* x Col-0 F1 (i.e. *GNL1/gnl1*) seedlings to provide the native, BFA-resistant GNL1 function, GNL1^sens^-YFP underwent only a very limited relocation to the PM following a BFA treatment (*Figure 5D and E*), indicating a selectivity of this process to GN. On the other hand, GNL1^sens^-YFP secreted to the PM strongly in homozygous *gnl1* mutant background (*Figure 5—figure supplement 1D*). This, however, was a manifestation of a non-specific disruption of the endomembrane system in the sensitized *gnl1* background, since a similar BFA-induced PM relocation of the TGN-localized VACUOLAR H^+^ ATPASE a1 subunit (VHAa1; *Dettmer et al., 2006*) was observed in *gnl1*, but not in the wild type (*Figure 5—figure supplement 1E*). The limited PM relocation of GNL1^sens^-YFP, as compared with GN-GFP, when evaluated in wild-type backgrounds, demonstrates the preferential association of GN to its specific, BFA-induced exocytic pathway.

We additionally analysed the BFA-induced localization patterns of the GN^fwr^ variant. Interestingly, GN^fwr^-GFP, which in native conditions is not present at the GA (*Figure 2*), became associated with the

TGN, now at the core of the BFA bodies, but did not accumulate at the PM (*Figure 5C and E*). As such, it appears that the *fwr* mutation affects not only the normal recruitment of GN to the GA, but also its association with the BFA-induced exocytic process. GN^{fwr} is presumably de novo recruited to the TGN following its inhibition with BFA. The exact mechanism behind these observations is presently unclear.

In summary, when inhibited by BFA, GN undergoes a specific exocytic relocation to the PM, likely bound to the surface of vesicles. GNL1 has a very limited affinity to this pathway. This process is distinct from a general relocation of endomembrane components caused by BFA in sensitized genetic backgrounds. Although GN secretion is most likely only a BFA-induced process, BFA-inhibited GN may be recruited to these exocytic vesicles by molecular interactions which distinguish it from GNL1. Thus, the secretory pathway leading to the PM may contain unknown molecular components required for the GN-specific activity in developmental patterning. These observations are in line with the notion that the GN-specific activity is mediated from the cell periphery.

## GN-GNL1 chimeras suggest functionally overlapping GN-specific features in all GN domains

The comparison of GN and GNL1 presented above uncovers a localization of GN to unknown structures at or in close proximity to the PM, and a BFA-induced association of GN with unknown molecular components present on vesicles trafficking to the PM. Both these phenomena are specific to GN when compared with its homologue GNL1. We next investigated where in the GN protein are found sequences giving GN its distinct characteristics, manifested by these cell biological phenomena, and ultimately, responsible for GN's unique function in development.

The large ARF-GEFs of GBF1 class, as well as the related BIG ARF-GEFs, are composed of a similar set of domains. At the N-terminus are the DCB (dimerization and cyclophilin binding) and HUS (homology upstream of SEC7), followed by the SEC7 domain, and finally by HDS (homology downstream of SEC7) at the C-terminus, of which there are 3 in GBF1 and 4 in BIG class ARF-GEFs (*Wright et al., 2014*). The SEC7 domain engages directly with an ARF substrate and catalyses the exchange of GDP to GTP (*Mossessova et al., 1998*; *Cherfils et al., 1998*), leading to membrane docking and activation of the ARF. The functions of the non-catalytic DCB, HUS, and HDS domains are partially characterized. Studies of BIG and GBF1 class ARF-GEFs show that these domains function in membrane association of the large ARF-GEFs, and in allosteric regulations of the catalytic activity of the SEC7 domain (*Richardson et al., 2012*; *Richardson et al., 2016*; *Galindo et al., 2016*; *McDonold and Fromme, 2014*; *Halaby and Fromme, 2018*; *Gustafson and Fromme, 2017*; *Nawrotek et al., 2016*; *Meissner et al., 2018*). These functions often depend on interactions of the non-catalytic domains with membranes, either direct, or through binding to activated ARF, ARF-LIKE, and Rab GTPases. In this sense, ARF-GEFs are not only activators, but also effectors of small GTPases, creating positive feedback loops, and cascades, in small GTPase networks (*Stalder and Antonny, 2013*; *Richardson and Fromme, 2012*; *Lowery et al., 2013*). In GN, the DCB domain is known to be responsible for homodimerization and participates in intramolecular interactions (*Grebe et al., 2000*; *Anders et al., 2008*; *Brumm et al., 2022*).

The difference in the molecular functions of GN and GNL1, ultimately expressed in their functions on the organismal scale, and evidenced by a lack of *gn* mutant rescue by GNL1 expression, by the association of GN to the PM-localized structures, and by the preferential association of GN to the BFA-induced secretory process, must be, in the most basic sense, reflected in the differences in the amino acid sequences of these two ARF-GEFs. To identify the determinants of this special GN function within the GN protein sequence, we cloned a series of chimeric GBF1 class ARF-GEFs composed of domains originated from GN and GNL1, and tested their activity in the molecular function of GN. We divided the ARF-GEFs into three components: (i) the N-terminal regulatory DCB-HUS domains, (ii) the catalytic SEC7 domain, and (iii) C-terminal regulatory HDS1-3 domains. While the three HDS domains are likely not identical in activities, at this stage we analysed their contribution as a single entity. The resulting chimeras are named by three-letter acronyms indicating the subsequent components of GN (G) and GNL1 (L) origin (*Figure 6A*). Of the possible chimeras, we generated all except LLG. Constructs expressing GFP fusions of the chimeras, controlled by the *GN* promoter, were transformed into heterozygous *gn^s* plants to test their ability to replace the function of GN. Homozygous rescued mutants were isolated where possible; otherwise segregating *gn^s* progenies were analysed.

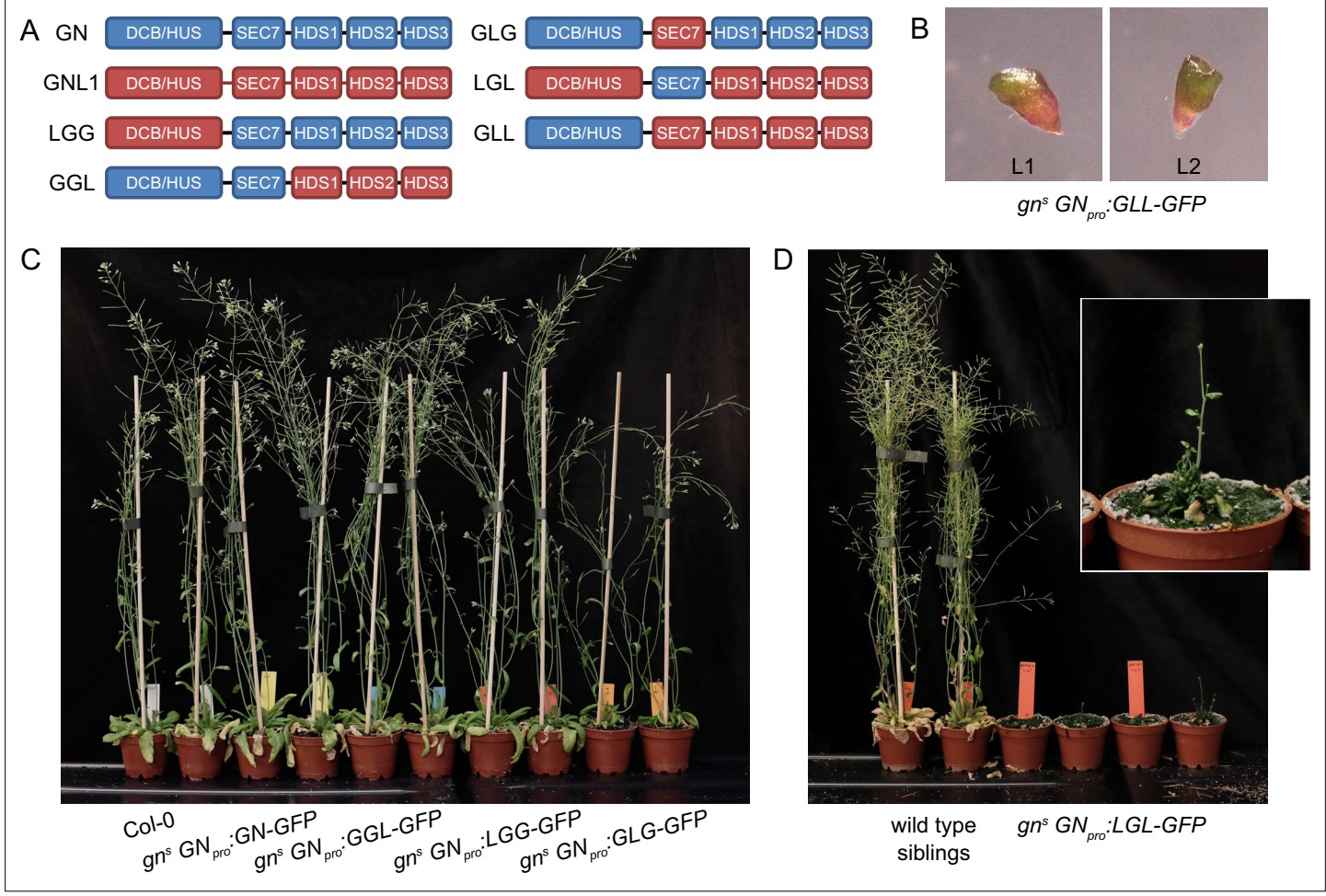

**Figure 6.** GNOM - GNOM-LIKE1 chimeric ARF-GEFs. (**A**) Schematic representation of the domain composition of GN, GNL1, and chimeric GBF1-type ARF-GEFs. An LLG chimera was not cloned. (**B**) No complementation of *gn^s* by the expression of GLL-GFP. (**C**) Phenotypes of adult *gn^s* mutants complemented by the expression of GN-GFP, GGL-GFP, LGG-GFP, and GLG-GFP. Expression of GGL-GFP and LGG-GFP fully complements the mutant phenotype, while plants expressing GLG-GFP develop smaller rosettes, fewer branches, and have limited fertility. (**D**) Phenotypes of adult *gn^s* mutants complemented by the expression of LGL-GFP. Expression of LGL-GFP provides only a modest level of complementation. Adults are characterized by limited growth, a compact rosette, and if bolting, are completely infertile. Wild-type siblings from a population segregating for *gn^s* are shown as control. Photographs in (**C**) and (**D**) show two independent transgenic lines expressing each construct.

The online version of this article includes the following figure supplement(s) for figure 6:

**Figure supplement 1.** Additional data related to GNOM - GNOM-LIKE1 chimeras.

All phenotypes described in the following were observed in two independent transgenic lines for each chimera (*Figure 7—figure supplement 2*; Materials and methods). *gn^s GN_pro:GN-GFP* and *gn^s GN_pro:GNL1-GFP* lines (*Figure 1—figure supplement 1*) constitute controls for this experiment. The summary of all described phenotypes is given in *Table 1*.

First, we consider in isolation the functions of DCB/HUS and HDS domains. The LGG chimera, consisting of the DCB/HUS domain of GNL1 and the remaining domains of GN, was fully functional, as reflected by normal phenotypes of the complemented mutants both at seedling and adult stages (*Figures 6C, 7A and B*). Root hair positioning at basal cell sides was also normal (*Figure 1—figure supplement 1D*). Conversely, a GLL chimera, where only the DCB/HUS domain originates from GN, did not exhibit any GN function, as *gn^s* seedlings expressing this chimera retained the typical *gn* phenotypes (*Figure 6B*). These observations indicate that differences between the DCB/HUS domains of GN and its homologue GNL1 are not responsible for the specific molecular action of GN.

Next, we consider the HDS domains. The GGL chimera, where the HDS domains of GN are replaced with those of GNL1, was almost fully functional: While the adult plants developed normally (*Figure 6C*),

Table 1. Summary of phenotypes of *gn*ˢ expressing ARF-GEF variants.

| ARF-GEF | Overall degree of GN function | Adult phenotype | Seedling growth rate | Root waving | Root hair positioning | Apical patterning | Etiolated seedling tropism | Apical hooks |
|---|---|---|---|---|---|---|---|---|
| GN (GGG) | complete | normal | normal | normal | normal | normal | normal | normal |
| LGG | complete | normal | normal | normal | normal | normal | normal | normal |
| GGL | very high | normal | normal | decreased | not analyzed | normal | normal | normal |
| GLG | moderate | decreased growth and fertility | partially decreased | decreased | not analyzed | ~50% single or fused cotyledons | variable growth directions | pen apical hooks |
| LGL | moderate to low | very small, infertile | strongly decreased | absent | not analyzed | ~50% single or fused cotyledons | variable growth directions | pen apical hooks |
| GLL | none | - | identical to *gn* | - | - | identical to *gn* | - | - |
| GNL1 (LLL) | none | - | identical to *gn* | - | - | identical to *gn* | - | - |
| LLG | not cloned | | | | | | | |

the seedlings, which were of normal size, exhibited a slight alteration in root growth pattern, characterized by a visibly more straight growth (*Figure 7A*). When grown in Petri dishes tilted back from the vertical, which in the wild type induces a wavy pattern of root growth, this straight growth was well expressed by the partial loss of root waving (*Figure 7B*). This phenotype may be caused by a slight defect in polar auxin transport, since low doses of auxin transport inhibitor N-1-naphthylphthalamic acid (NPA) produce similar effects (*Figure 7—figure supplement 1*). Overall, however, the phenotype of *gn*ˢ complemented by the expression of GGL indicates only a minor contribution of the HDS1-3 variants of GN, compared with those present in GNL1, to the function of GN.

Interestingly, however, when the DCB/HUS and the HDS domains of GN were simultaneously replaced by their counterparts from GNL1 in the LGL chimera, the GN-specific function was significantly affected. The adult *gn*ˢ *GN*ₚᵣₒ*:LGL-GFP* plants developed very slowly, and were characterized by small and compact rosettes of leaves. The individuals which did bolt, grew short stems with floral development arrested at an early stage (*Figure 6D*). Left to grow for several weeks after their wild-type siblings completed development, the plants grew numerous stems which remained short (*Figure 6—figure supplement 1A*). Ultimately, the plants were completely infertile, and the lines were maintained as *gn*ˢ +/-. Seedlings of *gn*ˢ *GN*ₚᵣₒ*:LGL-GFP* developed very short roots (*Figure 7C*), did not exhibit any root waving (*Figure 7D*), and in up to 50% of cases possessed single, or partially fused, cotyledons (*Figure 7C and E*), an embryonic defect typical to *gn* mutants (*Shevell et al., 1994*; *Geldner et al., 2004*). Dark-grown seedlings were clearly agravitropic (*Figure 7F* and *Figure 6—figure supplement 1C*) and their apical hooks were open at 75 hr of in vitro growth (*Figure 7G*). These various deficiencies demonstrate that the DCB/HUS and HDS domains of GN, compared with their counterparts in GNL1, do, in fact, possess features required for the specific molecular function of GN. That the function of GN becomes defective in result of their simultaneous absence, but not when lost individually, suggests that the molecular function of GN may be mediated through multiple and partially redundant molecular interactions in which these regulatory domains are involved.

Finally, we analysed the contribution of the catalytic SEC7 domain of GN. *gn*ˢ complemented by GLG, a chimera where the SEC7 domain of GN is replaced by its counterpart from GNL1, provided a partial, but significant, level of function. Adult plants developed clearly better than plants expressing LGL, but visibly weaker than plants expressing GN, GGL, or LGG. Compared with the latter, they were characterized by smaller rosettes, and by the presence of fewer stems, which, however, were long (*Figure 6C* and *Figure 6—figure supplement 1B*). The fertility of *gn*ˢ *GN*ₚᵣₒ*:GLG-GFP* plants was limited.

A comparison of seedling phenotypes between *gn*ˢ complemented by GLG and by LGL was the most salient observation in our study of GN-GNL1 chimeras. These two chimeras express mutually exclusive domains of GN, and yet, the phenotypes resulting from *gn*ˢ complementation were, in the qualitative sense, the same. Specifically, similarly to *gn*ˢ *GN*ₚᵣₒ*:LGL-GFP*, *gn*ˢ *GN*ₚᵣₒ*:GLG-GFP* seedlings had partially reduced root growth (*Figure 7C*), a deficiency in the root waving response (*Figure 7D*),

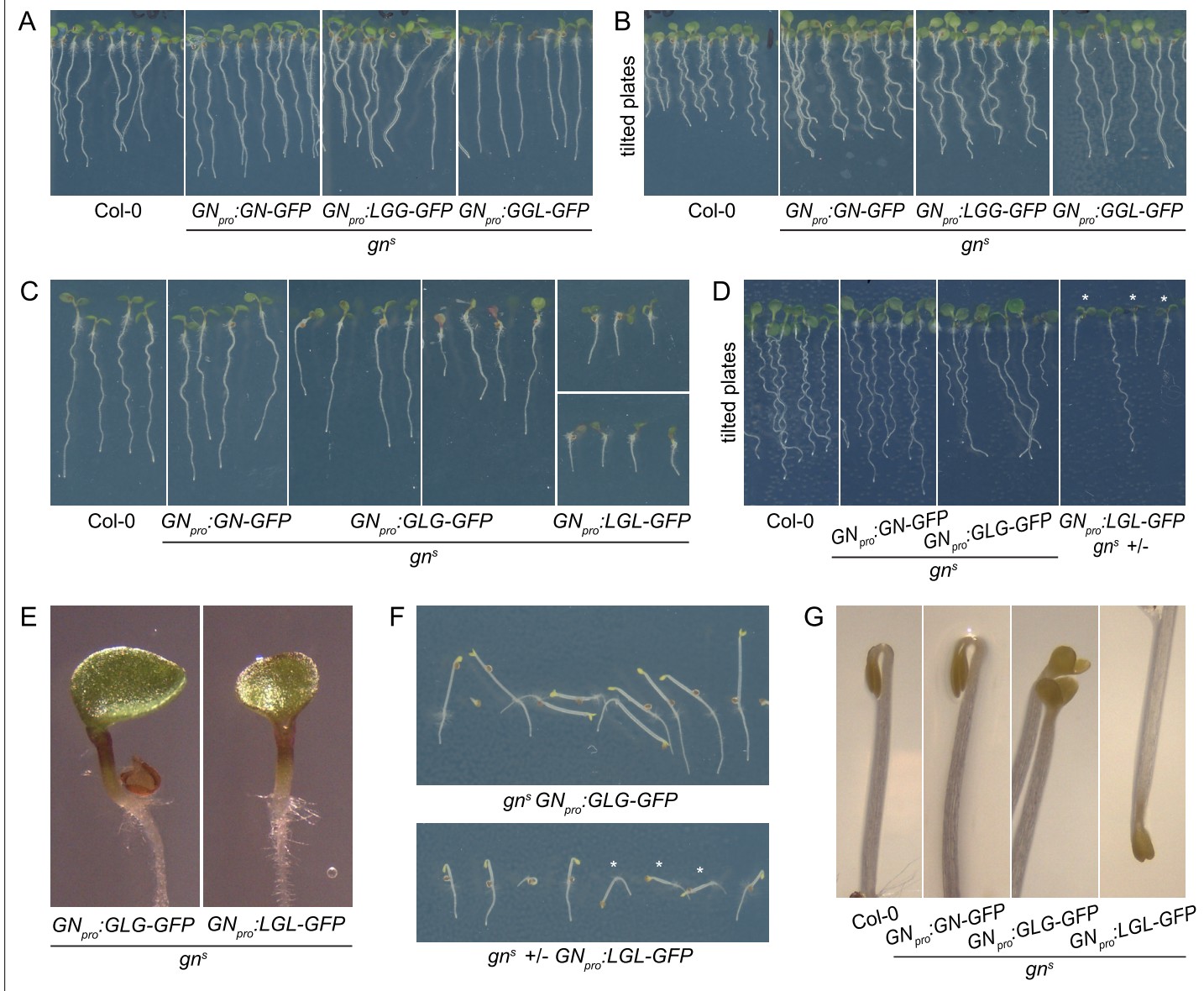

**Figure 7.** Seedling phenotypes of *gn*ˢ mutants complemented with GGL-GFP, LGG-GFP, GLG-GFP, and LGL-GFP. (**A**) Seedlings of *gn*ˢ mutants expressing LGG-GFP and GGL-GFP. Expression of LGG-GFP fully complements the mutant phenotype, while seedlings expressing GGL-GFP are characterized by roots growing more straight than controls. (**B**) Seedlings of *gn*ˢ mutants expressing GGL-GFP and LGG-GFP growing on agar plates tilted back from the vertical position. The root waving phenotype is normal in seedlings expressing LGG-GFP, while seedlings expressing GGL-GFP exhibit decreased waving. (**C**) Seedlings of *gn*ˢ mutants expressing GLG-GFP and LGL-GFP. Seedlings expressing GLG-GFP have moderately and variably decreased root lengths, while seedlings expressing LGL-GFP have strongly decreased root lengths. Both lines exhibit single or partially fused cotyledons in up to 50% of seedlings (GLG-GFP: right panel; LGL-GFP: bottom panel). (**D**) Seedlings of *gn*ˢ mutants expressing GLG-GFP and LGL-GFP growing on agar plates tilted back from the vertical position. The root waving phenotype is less expressed in mutants complemented with GLG-GFP, and completely absent in mutants complemented with LGL-GFP. The rightmost panel shows a population segregating for *gn*ˢ, complemented homozygous mutants are marked by asterisks. (**E**) Detail of single cotyledon phenotypes of *gn*ˢ mutants complemented by the expression of GLG-GFP and LGL-GFP. (**F**) Etiolated seedlings of *gn*ˢ mutants expressing GLG-GFP and LGL-GFP exhibit agravitropic growth. The bottom panel shows a population segregating for *gn*ˢ, complemented homozygous mutants are marked by asterisks. (**G**) Etiolated seedlings of *gn*ˢ mutants expressing GLG-GFP and LGL-GFP exhibit open apical hooks at 75 hr of in vitro growth.

The online version of this article includes the following figure supplement(s) for figure 7:

**Figure supplement 1.** The effect of NPA on root waving.

**Figure supplement 2.** Additional data on seedling phenotypes of *gn*ˢ mutants complemented with GN-GNL1 chimeras.

exhibited up to 50% of seedlings with partially or completely fused cotyledons (*Figure 7C and E*), had agravitropic etiolated seedlings (*Figure 7F* and *Figure 6—figure supplement 1C*), and open apical hooks at 75 hr of development in in vitro cultures (*Figure 7G*).

This comparison of chimeras with mutually exclusive domain composition shows clearly that the unique molecular function of GN cannot be ascribed to any single domain of the GN protein. It is likely that multiple interactions of GN DCB/HUS, SEC7, and HDS domains, in allosteric regulations within single GN proteins, during dimerization, as well as between GN domains and their unknown external interactors, including small GTPases, may all contribute, in a partly redundant, quantitative sense, to the specific molecular function of GN, distinguishing this ARF-GEF from GNL1.

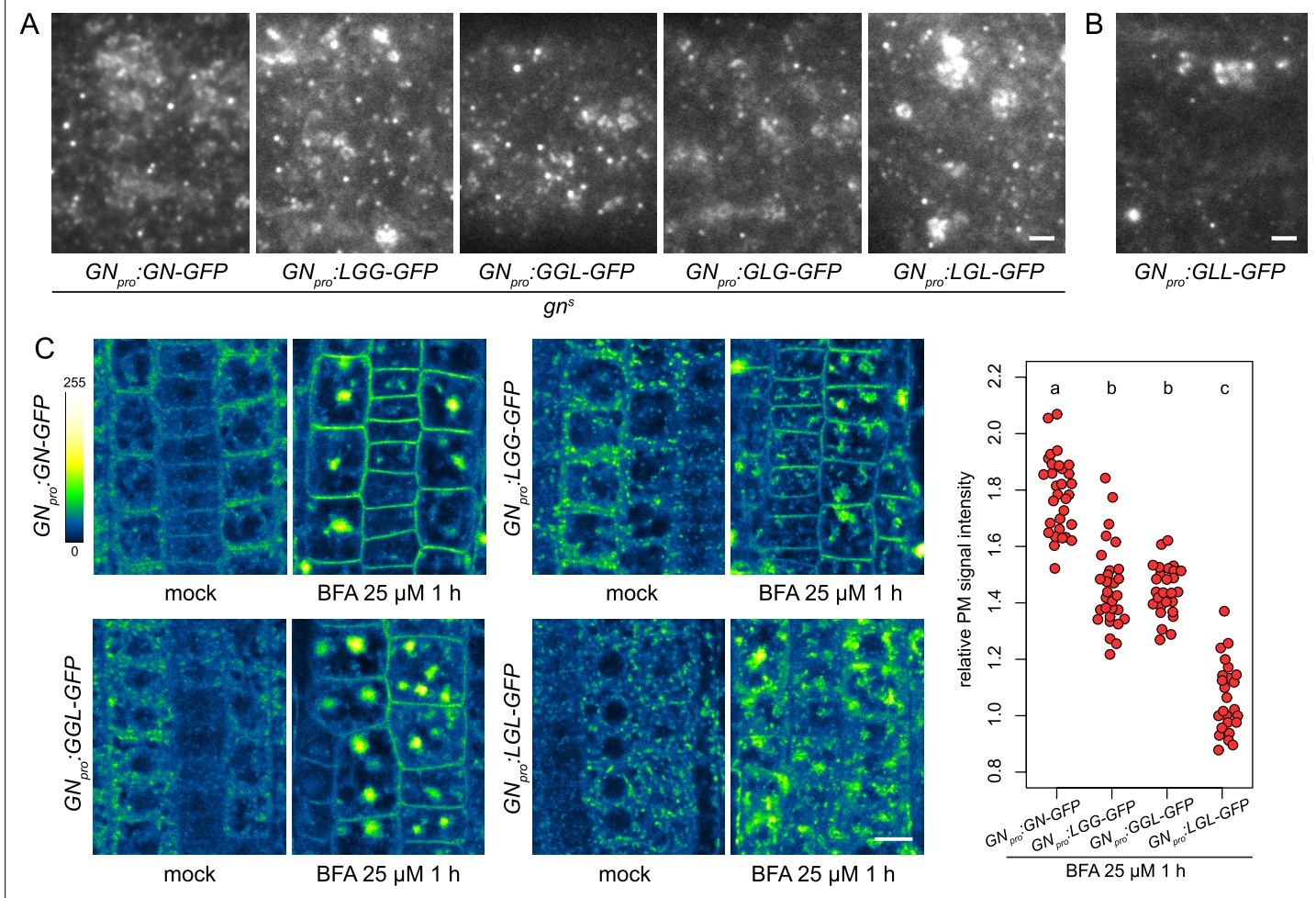

**Figure 8.** Subcellular localization and BFA-induced exocytosis of ARF-GEF chimeras. (**A**) TIRF images of GN-GFP, LGG-GFP, GGL-GFP, GLG-GFP and LGL-GFP in hypocotyls of etiolated seedlings of complemented *gn$^s$* mutants. All complementing chimeras localize to the GN-specific structures at the PM, and to GA. Scale bar - 2 µm. (**B**) TIRF image of GLL-GFP in wild-type etiolated seedling hypocotyl. GLL-GFP can be identified at GN-specific structures at the PM, and the GA. Scale bar - 2 µm. (**C**) CLSM images of seedling RAM epidermis of GN-GFP and chimeras possessing the BFA-sensitive SEC7 domain of GN, expressed in wild-type background, following a treatment with BFA at 25 µM for 1 hr. LGG-GFP and GGL-GFP localize to the PM after a BFA treatment, while LGL-GFP does not. Scale bar - 10 µm. Graph shows a quantification of PM signal intensities relative to total RAM signals following treatments with BFA at 25 µM for 1 hr from a representative experiment. A sum of data from two transgenic lines expressing each fluorescent reporter is shown. GN-GFP: 1.79±0.13 (mean ± s.d.), n=32; LGG-GFP: 1.46±0.15, n=29; GGL-GFP: 1.46±0.09, n=27; LGL-GFP: 1.07±0.12, n=26. Values were compared using One-way ANOVA (p<0.0001) with post-hoc Tukey HSD test, groups of significantly different values are indicated.

The online version of this article includes the following figure supplement(s) for figure 8:

**Figure supplement 1.** Additional data on subcellular localization and BFA-induced exocytosis of ARF-GEF chimeras.

## Subcellular localization and BFA-induced exocytosis of GN-GNL1 chimeras

To complete the analysis of the GN-GNL1 chimeras, we assessed them in the context of the GN activity at the newly identified structures at the PM (*Figure 1*), and of the GN-specific BFA-induced exocytic process (*Figure 5*). When etiolated hypocotyls of *gn$^s$* complemented by any of the functional chimeras, that is, LGG-GFP, GGL-GFP, GLG-GFP, and LGL-GFP, were analysed by TIRF microscopy, all these chimeric ARF-GEFs localized to the GA as well as the punctate structures at the PM, identical in appearance to those where GN-GFP localizes (*Figure 8A* and *Figure 8—figure supplement 1A*). The non-functional chimera GLL-GFP, expressed in the wild type, could be seen at the GA and relatively sparsely at GN-specific PM structures, as well (*Figure 8B*). The recruitment of the chimeric ARF-GEFs to these structures at the PM, mediated through any domain of GN, supports our conclusion of their role in the specific developmental function of GN, but the localization of GLL-GFP indicates that localization alone may not be sufficient to carry out GN function.

Next, we tested whether chimeric ARF-GEFs are recruited into the GN-specific, BFA-induced exocytic pathway. We used chimera-GFP reporters expressed in the wild type, rather than the variably rescued mutants, to obtain more directly comparable results. In these lines, the chimeras were not reliably observed by CLSM at the PMs of seedling RAMs in undisturbed conditions. This may be due to low signal intensity at the PM relative to the cytosolic and GA-localized signals, possibly resulting from competition with the native GN, which presumably recruits to the PM sites of action more effectively. Overall, in the course of experiments presented in this study, the localization of GN to the PM was most reliably observed with TIRF microscopy, rather than by CLSM; as discussed above, somewhat analogically, CCPs are always observed at the PM with TIRF microscopy, but the detection of fluorescent protein fusions of CLC at the PMs of RAM epidermis by CLSM is variable (e.g. *Adamowski et al., 2018*, *Adamowski et al., 2021b*).

Predictably, due to their resistance to BFA, chimeras with the SEC7 domain of GNL1, GLL-GFP and GLG-GFP, remained at the BFA body periphery where GAs are present, and did not undergo BFA-induced relocations (*Figure 8—figure supplement 1B*), similarly to the native, BFA-resistant GNL1 (*Figure 5—figure supplement 1C*). A comparison of BFA-induced relocation to the PM was conducted between GN-GFP, LGG-GFP, GGL-GFP, and LGL-GFP, that is, chimeras with the BFA-sensitive SEC7 domain of GN. We found that the chimeras with relatively larger portions of the GN protein, that is, LGG-GFP and GGL-GFP, significantly relocated to the PM following BFA treatments, although to a degree measurably lower than GN-GFP (*Figure 8C* and *Figure 8—figure supplement 1C*). In turn, the chimera with a smallest contribution of GN sequence, LGL-GFP, did not exhibit any relocation, and likely remained at the GA (*Figure 8C* and *Figure 8—figure supplement 1C*). These observations further show the selective affinity of GN, compared with GNL1, to the BFA-induced exocytic process, and reflect the manner in which the domains of GN quantitatively contribute to the molecular character of this ARF-GEF.

## Discussion

The mechanism behind the unique developmental function of GN ARF-GEF has been a subject of intense research. The information obtained over the last decades significantly contributed to our understanding of how the function of the endomembrane system impacts on patterning and polarity of the whole plant body. In this study, we provide novel insights into this function, taking into consideration the molecular nature of GN as an ARF-GEF component acting in the endomembrane system, and taking advantage of GNL1, its close homologue in terms of sequence and structure, which does not share GN's developmental function.

With the use of GN-GNL1 chimeras, we attempted to identify specific domains of the GN protein responsible for its activity. This analysis indicated that all GN domains: the regulatory DCB/HUS and HDS domains as well as the catalytic SEC7 domain, contain unique sequences that contribute, in a somewhat redundant fashion, to placing GN in a specific molecular environment within the cell. The hypothetical molecular interactions collectively mediated by these GN-specific domains promote GN action at the cell periphery. This GN-specific mechanism may potentially involve endomembrane components present on exocytic vesicles, as is suggested by GN's affinity for the exocytic pathway induced by its inhibition with BFA. This artificially induced process of GN exocytic relocation could be

observed due to the incidental combination of BFA-resistant and BFA-sensitive ARF-GEF isoforms in the *A. thaliana* model system. Were GNL1 BFA-sensitive, the specific secretion of GN could not be pinpointed due to a general endomembrane system breakdown, and were BIG3 BFA-sensitive, no GN-specific exocytosis would occur due to complete BFA sensitivity of the TGN.

One central proposition of our work is that the developmental function of GN is mediated from the cell periphery, from rather stable structures of an unknown nature, which may be bound to, or localize in a close proximity to, the PM. This interpretation is based on the observation that the peripheral localization distinguishes functional GN-GFP reporters, as well as GN-GNL1 chimeras, from GNL1. Even more suggestively, GN^fwr, a mutant variant of GN exhibiting an almost full functionality in development, localizes to these structures and cannot be detected at the other detectable site of GN binding, the GA. This model is consistent with the observation that the ARF-GAP VAN3, which exhibits a function in auxin-mediated developmental patterning like GN, acts from the PM as well (*Naramoto and Kyozuka, 2018*), although in microscopy VAN3 appears as dense and very dynamic structures distinct from those that recruit GN (*Adamowski and Friml, 2021a*). While the PM localization of GN was reported previously, advances in live imaging lead us to conclude that the PM functions of both GN (this study) and VAN3 (*Adamowski and Friml, 2021a*) do not rely on the regulation of CME (*Naramoto et al., 2010*). In turn, the localization of GN to the GA, which, similarly, was clarified from the previous endosomal site of action thanks to technological advances (*Geldner et al., 2003*; *Naramoto et al., 2014*), is now proposed to represent only, or at least to a major degree, its contribution to the fundamental secretory activity, an ancestral role which it shares with GNL1 (*Richter et al., 2007*).

A direct action of GN at the cell periphery appears consistent with the rapid and sensitive manner in which its inhibition by BFA causes a loss of D6PK from its polar PM domain (*Barbosa et al., 2014*). With regard to the mechanism by which GN controls the polar distribution of PIN auxin transporters, our findings argue against a model where GN promotes an ARF-dependent formation of exocytic vesicles trafficking PINs from an intracellular compartment to the polar domain at the PM. Considering not only its site of action at the PM required for developmental patterning, but also the reported non-cell autonomous effect of GN on PIN polarization (*Wolters et al., 2011*), we favor an indirect model of GN action on PIN polarity, where its activity influences other, unknown polarity components, which in turn instruct the vesicular sorting or polar retention of PINs (*Glanc et al., 2021*). It is possible that the polarity determinants regulated by GN at the PM are in some way associated with the cell wall. This is hinted by the alteration of cell wall structure in *gn* mutants (*Shevell et al., 2000*; *Wachsman et al., 2020*), but also by the identification of a cellulose synthase mutant in the *regulator of pin polarity (repp)* forward genetic screen, where the loss of cellulose synthase action caused an ectopically expressed PIN1 to be sorted apically, rather than basally, in epidermal cells (*Feraru et al., 2011*). It could be conceived that GN-controlled polarity determinants within the cell wall, or associated with the cell wall from the PM, provide information orienting not only the polar domains of individual cells, but also, as an apoplastic continuum, coordinating symplast polarity patterns on the tissue scale (*Steinmann et al., 1999*). This enables the patterned development of organs, of tissues, prominently vasculature (*Verna et al., 2019*), and of the body plan as a whole (*Mayer et al., 1991*).

## Materials and methods

**Key resources table**

| Reagent type (species) or resource | Designation | Source or reference | Identifiers | Additional information |
|---|---|---|---|---|
| Biological sample (*Arabidopsis thaliana*) | Col-0 (Columbia) | Nottingham *Arabidopsis* Stock Centre (NASC) | N1092 | |
| Gene (*Arabidopsis thaliana*) | GNOM | The *Arabidopsis* Information Resource | AT1G13980 | |
| Gene (*Arabidopsis thaliana*) | GNOM-LIKE1 | The *Arabidopsis* Information Resource | AT5G39500 | |
| Genetic reagent (*Arabidopsis thaliana*) | gn^s | SALK T-DNA collection | SALK_103014 | |

### Plant material

The following previously described *A. thaliana* lines were used in this study: *gn^s* (SALK_103014; *Okumura et al., 2013*), *GN_{pro}:GN-GFP VAN3_{pro}:VAN3-mRFP*, *GN_{pro}:GN-GFP 35S_{pro}:CLC-mKO* (*Naramoto et al., 2010*), *CLC2_{pro}:CLC2-GFP* (*Konopka et al., 2008*), *LAT52_{pro}:TPLATE-GFP*

RPS5A$_{pro}$:AP2A1-TagRFP (*Gadeyne et al., 2014*), secRFP (NASC ID N799370; *Samalova et al., 2006*), big3 (SALK_044617; *Richter et al., 2014*), XVE»amiCHCa (*Adamowski et al., 2021b*), gnl1 GNL1$_{pro}$:GNL1$^{sens}$-YFP (*Richter et al., 2007*), VHAa1$_{pro}$:VHAa1-GFP (*Dettmer et al., 2006*), gnl1-2 (*Teh and Moore, 2007*). The lines generated as part of this study as listed in *Supplementary file 1a*.

## In vitro cultures of *Arabidopsis* seedlings

Seedlings were grown in in vitro cultures on half-strength Murashige and Skoog (½MS) medium of pH = 5.9 supplemented with 1% (w/v) sucrose and 0.8% (w/v) phytoagar at 21 °C in 16 hr light/8 hr dark cycles with Philips GreenPower LED as light source, using deep red (660 nm)/far red (720 nm)/blue (455 nm) combination, with a photon density of about 140 μmol/(m²s)+/-20%. Petri dishes for TIRF imaging in hypocotyls of etiolated seedlings, and for studies of development of etiolated seedlings, were initially exposed to light for several hours and then wrapped in aluminium foil.

## Chemical treatments

BFA (Sigma-Aldrich B7651) was solubilized in DMSO to 50 mM stock concentration and added to liquid ½MS media for treatments. Beta-estradiol (Sigma-Aldrich E8875) was solubilized in 100% ethanol to 5 mg/mL stock concentration and added to ½MS media during preparation of solid media to a final concentration of 2.5 μg/mL. Induction of XVE»amiCHCa was performed approximately 48 hr before CLSM imaging by transferring 3-day-old seedlings to media supplemented with beta-estradiol.

## Molecular cloning and generation of transgenic lines

All constructs were generated using the Gateway method (Invitrogen) and are listed in *Supplementary file 1c*. DNA sequences were amplified by PCR using iProof High Fidelity polymerase (Bio-Rad). Primers used for cloning are listed in *Supplementary file 1b*. GN-GNL1 chimeras were cloned using overlap extension PCR (fusion PCR) from cDNA templates. Fragment 1 (DCB-HUS) corresponds to nucleotides 1–1653 in both ARF-GEFs, fragment 2 (SEC7) to nucleotides 1654–2268 in both ARF-GEFs, and fragment 3 (HDS1-3) to nucleotides 2269–4329 and 2269–4353 in *GN* and *GNL1*, respectively. *GN, GNL1* (of sequences corresponding to the reference Araport11 genome) and chimeric *GGL, GLG, LGG, GLL, LGL* sequences without stop codons were introduced by Gateway BP Clonase into pDONR221 vectors. *GN* promoter region (nucleotides –2127 to –1; *Geldner et al., 2003*) was PCR-amplified and introduced by Gateway BP Clonase into pDONRP4P1r vector. Expression vectors were made by combining *GN$_{pro}$*, ARF-GEF coding sequences, and GFP/pDONRP2rP3, in pH7m34GW expression vector (*Karimi et al., 2002*) using Gateway LR Clonase II Plus. In result, all described constructs encode contain ARF-GEFs linked at their C-termini to GFP by a linker of amino acid sequence DPAFLYKVG. *FWR* mutation was introduced using overlap extension PCR (fusion PCR) from *GN* cDNA template by substitution of $^{3013}$TCT for $^{3013}$TTT. The rest of the procedure was analogical to that described above. gns +/-were transformed by the standard floral dip method, and T1 plants were selected on ½MS medium without sucrose supplemented with hygromycin. T1 plants were genotyped for gn$^s$ insertion using primers gn-SALK-F, gn-SALK-R, and LBb1.3, of which gn-SALK-R binds to the *GN* intron absent in the transgenes. Where gn$^s$ -/- were isolated in T1 (GN, LGG, GGL, GLG, and GN$^{fwr}$), between 3 and 8 gn$^s$ -/- lines of consistent complementation phenotype were initially obtained. Subsequently two representative lines were selected for detailed analyses. Where gn$^s$ -/- were not isolated in T1 (GNL1, GLL, LGL), multiple gns +/-were initially selected and T2 generation of three gns +/-lines with verified GFP signals were sown for further genotyping. No complementation (no gn$^s$ -/- individuals) by GNL1 or GLL was confirmed in these lines, and consistent sterile, dwarf adult phenotype of gn$^s$ -/- complemented with LGL was observed before a detailed characterization of two representative lines.

## Light microscopy

High-magnification images of seedlings developing in in vitro cultures were taken with Leica EZ4 HD stereomicroscope equipped with ×0.8–3.5 magnification lens, by an integrated 3 megapixel CMOS camera.

## Root hair positioning

Relative root hair positions were measured on light microscopic images using Fiji (https://imagej.net/Fiji). Distances of root hair outgrowths from apical (a) and basal (b) cell ends were measured and relative position calculated as b/(a+b).

## Confocal laser scanning microscopy

Four- to 5-day-old seedlings were used for live imaging with Zeiss 800 confocal laser scanning microscope with 10X0.45 air, 20X0.8 air, and 40X1.2 water lenses. Excitation wavelengths 488 nm and 561 nm. Detector type: two gallium arsenide phosphide photomultiplier tube detectors (GaAsP PMTs) with free choice of spectral range (emission ranges were selected to optimize detection while reducing background autofluorescence). Gain was set according to fluorescent protein expression levels in the range 650–800 V. Line averaging 2 X, offset 0. Z-stacks were captured with 0.6–1 µm spacing. Relative PM signal intensities of ARF-GEF-GFP fusions following BFA treatments were measured using Fiji (https://imagej.net/Fiji) as a ratio between mean grey values of a line of 5 pixel width drawn over multiple PMs, and of a rectangle covering the whole RAM surface visible in a CLSM image. Sample sizes (given in figure legends) and number of repetitions (2-4) were decided through experimenter's experience.

## Total internal reflection fluorescence microscopy

Early elongation zone of roots in excised ~1 cm long root tip fragments from 7-day-old seedlings, as well as 3- to 5-day-old etiolated wild-type hypocotyls and *gn* mutant seedlings, were used for TIRF imaging. Imaging was performed with Olympus IX83 TIRF microscope, using a 100X1.40 oil UAPON OTIRF lens with an additional ×1.6 magnification lens in the optical path. Excitation wavelengths 488 nm and 561 nm. Due to the nature of imaged tissues, laser illumination angle was adjusted individually to each sample to optimize detection of signals near cell surface with minimal background. Detector type: Hammamatsu ImagEM X2 EM-CCD C9100-13. EM gain was set in the range 200–400 V depending on fluorescent reporter expression levels. For single channel GFP imaging, a dichroic LM491 filter and a 525/45 bandpass filters were used. For two channel imaging, a quad line beamsplitter ZT405/488/561/640rpc (Chroma) was employed (reflected wavelengths 400–410 nm, 485–493 nm, 555–564 nm, 630–644 nm). Time lapses of 100 frames at 1 s intervals with exposure times of 200ms, or single snapshots of 200ms exposure, were taken, depending on the experiment. Two-channel imaging was performed sequentially with 200ms exposure times. Quantitative evaluations were performed using Fiji (https://imagej.net/Fiji). CLC2-GFP and TPLATE-GFP foci were counted in square regions of 36 µm$^2$ taken from the captured TIRF images or movies. Lifetimes of CLC2-GFP and TPLATE-GFP were measured in kymographs extracted from each captured TIRF movie. GN-GFP foci were counted in square regions of 110.25 µm$^2$ taken from the captured TIRF images or movies. Lateral displacement of GN-GFP signals over time was measured on maximum projections of kymographs. Sample sizes (given in figure legends) and number of repetitions (2-4) were decided through experimenter's experience.

## Accession numbers

Sequence data from this article can be found in the GenBank/EMBL libraries under the following accession numbers: GNOM (AT1G13980), GNOM-LIKE1 (AT5G39500), VAN3 (AT5G13300), BIG3 (AT1G01960), CLC2 (AT2G40060), TPLATE (AT3G01780), VHAa1 (AT2G28520).

# Acknowledgements

The authors would like to gratefully acknowledge Dr Xixi Zhang for cloning the *GNL1*/pDONR221 construct and for useful discussions.

## Additional information

### Funding

| Funder | Grant reference number | Author |
|---|---|---|
| H2020 European Research Council | Advanced Grant ETAP-742985 | Jiří Friml |
| Austrian Science Fund | I 3630-B25 | Jiří Friml |

The funders had no role in study design, data collection and interpretation, or the decision to submit the work for publication.

### Author contributions

Maciek Adamowski, Conceptualization, Data curation, Formal analysis, Validation, Investigation, Visualization, Methodology, Writing - original draft, Writing - review and editing; Ivana Matijević, Investigation; Jiří Friml, Supervision, Funding acquisition, Validation, Project administration, Writing - review and editing

### Author ORCIDs

Maciek Adamowski  http://orcid.org/0000-0001-6463-5257
Jiří Friml  http://orcid.org/0000-0002-8302-7596

### Decision letter and Author response

Decision letter https://doi.org/10.7554/eLife.68993.sa1
Author response https://doi.org/10.7554/eLife.68993.sa2

## Additional files

### Supplementary files

• Supplementary file 1. Additional materials and methods. (a) Lines generated as part of this study. (b) Primers used in this study. (c) Constructs generated in this study.

• Transparent reporting form

### Data availability

All data generated or analysed during this study are included in the manuscript and supporting files. No additional source files were provided.

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
