## [Editor Report]

Many plant developmental processes are mediated by gradients of the hormone auxin. GNOM is a vesicle trafficking component (an ARF-GEF) that is presumably involved in the polar transport of PINs, auxin transporters that help to establish auxin gradients. Here, the authors challenge the model that GNOM acts at intracellular compartments by providing convincing microscopy evidence that GNOM primarily acts at the cell cortex and/or the plasma membrane and they hypothesize that GNOM might establish/maintain some other unknown polarity cue at the plasma membrane. These valuable findings open new questions in the fields of plant developmental biology and plant cell biology.

---

## [Decision Letter]

**Decision letter after peer review:**

Thank you for submitting your article "Developmental patterning function of GNOM ARF-GEF mediated from the plasma membrane" for consideration by *eLife*. Your article has been reviewed by 3, one of whom is a member of our Board of Reviewing Editors, and the evaluation has been overseen by Jürgen Kleine-Vehn as the Senior Editor. The following individuals involved in review of your submission have agreed to reveal their identity: Enrico Scarpella (Reviewer #3).

Essential revisions:

1) To assess functionality of GN variants, the authors used the rescue of the seedling phenotype, including its lethality, of a strong gn mutant allele; however, those phenotypes only reflect near-complete loss of GN function. Indeed, different fluorescently tagged GNOM protein fusions have been generated in this manuscript and others, and all the fusions have been claimed to be fully functional based on their ability to rescue the seedling phenotype of a strong gnom mutant allele. Therefore, all the fusions seem to share "full functionality"; however, they also have different localization patterns. The most sensitive indicators of even partial loss of GNOM function are lateral root formation (Geldner 2004; Okumura 2013), root hair polar positioning (Fischer 2006) and vein network formation (Geldner 2004; Verna 2019); therefore, analysis of at least one of those diagnostic criteria should be included. At this moment, claims of full functionality of GNOM variants are not fully supported and prevent deep understanding: even gnom-fwr and gnom-B/E mutants look normal at the level that the authors have analyzed the functionality of GNOM variants in this manuscript.

2) Throughout the manuscript, localization of fluorescently tagged GNOM protein was determined in epidermal cells of either hypocotyl or roots. In these cells, GNOM function has never been shown to be required. Therefore, the biological relevance of GNOM localization in the imaged in this manuscript is unclear. This point could be addressed by:

2a. GN-GFP localization in the cells in which GNOM is known to function, i.e. those cells that participate in the processes that most suffer from loss of GNOM function.

2b. Document that indeed GNOM has functions in the cells that were used to determine GNOM localization and use such functions as diagnostic criteria of phenotypic rescue.

2c. Some combination of 2a and 2b; for example, examining GN-GFP localization during root hair positioning (Fischer et al. 2006 Curr Biol) and carefully quantifying the ability of different GN-GFP constructs used in this paper to complement this mild gn phenotype.

3) Steady-state localization of GNfwr-GFP at PM doesn't mean it's never intracellular (absence of evidence is not evidence of absence). This claim should be supported by quantitative analysis of multiple confocal stacks of whole cells (from multiple independent transgenic lines) to document whether any intracellular signal is detected and/or colocalization of PM puncta with a PM marker (e.g. PIP2A, LTI6b, WAVE131, WAVE138, or very short (<5min) FM4-64 treatment).

4) Both Reviewers 1 and 2 have requested further quantification of imaging experiments from multiple (preferably 3 or more) independent transgenic lines – please see their comments details on which experiments require further quantification and statistical analyses.

5) Both Reviewer 1 and 3 raised questions about the experiments presented in Figures 5 and 8. Please revise the rationale, results, and conclusions around these data to clarify the biological questions being addressed by these experiments.

*Reviewer #1 (Recommendations for the authors):*

In this paper, Adamowski and Friml present and investigate the observation that GNOM (GN), an ARF-GEF presumed act at the Golgi apparatus and/or endosomes, is localized to both intracellular structures (i.e. Golgi apparatus/TGN/endosomes) and the plasma membrane (PM). Although the observation that GN is partially localized to the PM is not new (Naramoto et al., 2014 Plant Cell), the authors describe this localization in more detail and hypothesize about what role GN might play at the PM. Importantly, they find that a GFP fusion to the protein encoded by a weak mutant allele of GN localizes primarily to the PM and can complement most gn knockout phenotypes, implying that most of GN's role is performed at the PM (or, at least, that other intracellular ARF-GEFs act redundantly with intracellular GN, but not PM GN). Domain-swap experiments between GN and GNL1 document that no single region/domain is responsible for the functional or localization differences between GN or GNL1. Their claims are supported by molecular complementation of different mutant backgrounds (usually with two independent transgenic lines) and by high quality live cell imaging experiments. Most of the presented evidence are qualitative observations and some experiments are appropriately quantified.

The authors assume that the unique molecular function of GN is related to this PM localization, and that any intracellular functions performed by GN are redundant with GNL1. This hypothesis separates GN function from a direct influence on PIN trafficking from the endomembrane system, and rather implicates GN in the establishment/maintenance of some other unknown polarity cue that will subsequently affect PIN trafficking/localization. As a result, these findings potentially have important implications for the study of auxin, and therefore plant growth and development. Clearly, many questions remain about what polarity cues GN might influence and the mechanisms by which GN exerts its influence at the PM.

Overall, this manuscript documents and investigates an interesting observation that GN is partially localized to the PM. These results have important implications for plant biologists studying auxin, intracellular trafficking, and plant cell polarity. Below, I have raised several points that the authors should address to improve the connection between author claims and evidence (items 1-4, 6), to provide appropriate quantification (item 5), to clarify the methods (items 7 and 8).

1. Steady-state localization of GNfewerroots-GFP at PM doesn't mean it's never in the Golgi/TGN (absence of evidence is not evidence of absence). At the very least, this claim should be supported by quantitative analysis of multiple confocal stacks of whole cells (from multiple independent transgenic lines) to document whether any intracellular signal is detected.

2. I am confused by the notion of GN being "specifically exocytosed" (line 19) "anterograde secretory movement of GN through the endomembrane system" (line 349), "GN-exocytosis" and the "BFA-induced exocytic process"; isn't GN a soluble protein? How is it "in" the endomembrane system? Is it not just peripherally associating with membranes? I think the text explaining the rationale, results, and conclusions around figures 5, 8, and S3 need to be substantially reworded to clarify this. I think it's very important to clarify whether GN localizes to the PM via vesicle trafficking (i.e. "secretion" or "exocytosis") or via peripheral association with PM proteins (which may be the thing that is actually mislocalized during BFA treatment of big3 mutants).

3. It is unclear why the authors chose to use the GNpro:GN-GFP VAN3pro:VAN3-mRFP line for later experiments (Figure 5), particularly since it shows far less association with the PM (Figure 5A vs Figure 1A) the GNpro:GN-GFP line. Please clarify.

4. How many unique transformation lines were observed to draw the conclusion that GN-GFP is "always" partly localized to the PM, and how many GNL1-GFP lines were observed to support the conclusion that GNL1 is "never" found at the PM? I see evidence of only two GN-GFP lines in Figure S1, but it is unclear whether more than one line was used for imaging. Similarly, for the chimeric complementation experiments, the authors state that they analyzed two independent transgenic lines; three is usually the absolute minimum number, but even so, the data presented in Figure 6-8 seem to be from only one line. Please provide evaluations of multiple independent transgenic lines in supplemental data.

5. Many of the claims are qualitative and could easily be supported by quantitative data:

­ That GN localizes to "often stable" punctae in the PM – what is the average (and range) of the size of these particles, their density in the PM, their lifetime, their speed of movement, and their range of movement?

­– That GN signal is enhanced at the PM after BFA treatment (line 163): what is the change in mean signal intensity? How are these particles different from those observed under control conditions: what is the average (and range) of the size of these particles, their density in the PM, their lifetime, their speed of movement, and their range of movement?

– That gn knockouts expressing GNfwr are indistinguishable from gn knockouts expressing GN-GFP, which are indistinguishable from wild type: please quantify root length and test for statistically significant differences between multiple independent transformants and wild type.

– That GNfwr-GFP behaves in a manner indistinguishable from GN-GFP at the PM – what is the average (and range) of the size of these particles, their density in the PM, their lifetime, their speed of movement, and their range of movement? Are there any statistically significant differences between these metrics for GNfwr-GFP compared to GN-GFP?

– That GN-GFP is not colocalized with CME machinery (line 238): please quantify colocalization, and include an appropriate control (e.g. colocalization of 2 CME components)

– That bulk secretion is unaffected by loss of GN (line 274): please use a quantitative measure of secretion, for example, ratiometric secGFP (Samalova et al. 2015, Traffic) and quantify using both fluorescence ratios and Western blots. There seems to be a lot of signal aggregates in the gn seedling, which would actually suggest that secretion is affected. Please also include a wild type control at a similar developmental stage.

– The histogram in Figure 3D would be more appropriate as a violin plot. Please also use an appropriate statistical test to determine whether "event lifetime" is "the same as in wild type controls" (line 266) for both CME markers.

6. References to Adamowski and Friml 2021a (submitted), Adamowski and Friml 2021b (submitted), Adamowski and Friml 2021c (in preparation), and Adamowski et al., 2021 (in preparation) must be removed or replaced with references to published articles or published preprints. Any claims supported by these citations should be supported by references to published articles or published preprints; otherwise, data should be presented in this manuscript to support the claims, or the claims should be removed from the manuscript.

7. Many essential details are missing from the methods section and must be added:

– Please provide catalogue numbers for critical chemicals (e.g. BFA, β-estradiol).

– Light microscopy: details of the objective and camera used for imaging seedlings.

– CLSM: NA for the objectives used, excitation and emission wavelengths, z-spacing for any z-stacks, details on the detector type and detector settings used for image acquisition (e.g. PMT type with info on gain, offset, scan frequency, any line/frame averaging performed during collection).

– TIRFM: NA of objective, angle of illumination, excitation and emission wavelengths, details on the detector type and detector settings used for image acquisition (e.g. EMCCD or sCMOS type and settings).

8. Finally, please check the name of the gn T-DNA allele on line 136; Salk_103114 directed me here (https://abrc.osu.edu/stocks/number/SALK_103114) which is a T-DNA line affecting CESA2 and/or pre-tRNA.

*Reviewer #2 (Recommendations for the authors):*

The manuscript entitled "Developmental patterning function of GNOM (GN) ARF-GEF mediated from the plasma membrane" by Adamowski and Friml elaborated the analyses of GNOM localization at the plasma membrane. By using the TIRF microscopy, GN was determined to accumulate to unknown, relative static puncta at the plasma membrane. The authors further showed that the CME and bulk secretion processes appeared normally in gn mutants. Additionally, through domain exchange experiments, chimeric ARF-GEFs of GN-GNL1 were evaluated for their biological functions in vivo, leading to the conclusion that all domains of GN contribute to GN function.

The ARF-GEF GNOM has been known for its predominant localization at the Golgi apparatus to regulate secretion/recycling, thereby promote the polarization of PIN proteins at the plasma membrane in plant cells. Although a few previous publications reported the presence and function of GNOM at the plasma membrane (PM) (Naramoto 2010 and Okumura 2013), this study provides more in-depth cell biological analyses and established the localization pattern and dynamics of GN at the PM. These discoveries bring new insight into how GN may promote PIN polarization at differential subcellular locations, in particular the underexplored contribution from the plasma membrane pool. I am convinced that GN is partially localized to the plasma membrane beside the previously established localization at the Golgi but have the concern that the claim of "PM is the major place of GN action responsible for its developmental function" might be an overstatement.

1. The main basis for the authors' claim was (1) GN-GFP is partially localized to the PM as described in this story and previously in Naramoto 2010. The enrichment of GN-GFP at the PM was elevated by BFA treatment. (2) the localization of a GN mutant variant, GNfwr, is predominant at the plasma membrane. GNfwr contains a point mutation in one of the HDS domains that was screened for the mutant phenotype in lateral root initiation and abnormal patterning of auxin signaling in root development. Although GNfwr can largely complement the growth phenotypes of loss-of-function gn mutants, I doubt whether this provides the most solid evidence to make the conclusion. In any case, it remains unclear about the nature of the GNfwr defects and how GNfwr becomes more PM-localized. Was this due to interfered protein-protein or protein-membrane interactions, or due to the abnormal membrane trafficking when GNfwr is not able to localize to the Golgi? To make the experimental data more convincing, I would create situations that the wild-type GN protein can be functionally tested at the PM solely (without Golgi association) or at the Golgi solely (without PM association) in gn mutants. The localization and function of GNfwr might be too complicated to be heavily based on.

2. I wonder whether there is a way to better present the punctate structures that are indeed localized to or tightly associated with the plasma membrane but not underneath the membrane and in the cytoplasm (the TIRF images in Figure 1). Do these dots co-localize with the FM4-64 dye after a short-time treatment, or do they overlap with any of those Wave endosomal markers?

*Reviewer #3 (Recommendations for the authors):*

The cellular localization of the GNOM protein of Arabidopsis and the biological relevance of such localization has been object of interest for the past 20+ years. During this time, our knowledge of where in the cell the GNOM protein is and acts has continually been refined. In this manuscript, Adamowski and Friml continue to address this question and now suggest that the main site of action of the GNOM protein is not any of the intracellular membrane compartments, as previously thought, but the plasma membrane.

The gnom mutant is arguably the most severe patterning mutant in Arabidopsis – in the most extreme case, gnom mutant seedlings are nothing more than spheres seemingly lacking any polarity. As such, GNOM function is key to plant development, and understanding the cellular site of action of the GNOM protein is certainly one important step toward understanding such key function. Therefore, this study has the potential to advance and deepen our knowledge of how plants develop; however, for the following reasons the potential of the study is not yet fully realized.

1. In this manuscript, functionality of GNOM variants was assessed by their ability to rescue the seedling phenotype of a strong gnom mutant allele, including its seedling lethality. However, the most sensitive indicators of even partial loss of GNOM function are lateral root formation (Geldner 2004; Okumura 2013) and vein network formation (Geldner 2004; Verna 2019); therefore, analysis of those two diagnostic criteria – with which the authors are familiar – should be included. At this moment, claims of full functionality of GNOM variants are not fully supported and prevent deep understanding: even gnom-fwr and gnom-B/E mutants look normal at the level with which the authors have analyzed the functionality of GNOM variants in this manuscript.

2. Throughout the manuscript, localization of fluorescently tagged GNOM protein was determined in epidermal cells of either hypocotyl or roots. In those very epidermal cells GNOM function has never been shown to be required. Therefore, the biological relevance of GNOM localization in the epidermal cells of root or hypocotyl imaged in this manuscript is unclear. Localization should be determined in the cells in which GNOM is known to function, i.e. those cells that participate in the processes that most suffer from loss of GNOM function. I very well realize that because of technical limitations this is not an easy point to address, but in the absence of such evidence the biological relevance of the localization patterns shown in this manuscript is unclear at best. Alternatively, the authors could show that indeed GNOM has functions in the cells that were used to determine GNOM localization and use such functions as diagnostic criteria of phenotypic rescue (see point 1 above).

3. Different fluorescently tagged GNOM protein fusions have been generated in this manuscript and in previous ones, and all the fusions have been claimed to be fully functional based on their ability to rescue the seedling phenotype of a strong gnom mutant allele, including its seedling lethality. Therefore, all the fusions seem to share full functionality; however, they also seem to have different localization patterns. It's now important to evaluate claims of full functionality by assessing the ability of all the fusions to rescue the phenotype, including the lateral root and vascular network phenotypes, of the same gnom mutant allele: if some of those fusions were not localized to the plasma membrane and yet rescued all the phenotypes of a strong gnom mutant allele, including the lateral root and vein network phenotypes, plasma-membrane-localization of GNOM would not be relevant. Alternatively, it's possible that different developmental functions depend – to varying degrees – on GNOM localization to different cellular compartments.

4. In Figure 6, the authors investigate the question what domains in the GNOM protein are responsible for the functions that are unique to this protein and not to the GNOM-LIKE1 protein. To address this question, the authors create protein chimeras between GNOM and GNOM-LIKE1, and assess their ability to rescue the seedling phenotype of a strong gnom mutant allele, including its seedling lethality. This is an exciting approach, but also a preliminary one: not only should the phenotypic rescue include more sensitive diagnostic criteria (see point 1 above), but the number of chimeras generated is preliminary – for example, the study lumps together all the three HDS domains, as if they may not have separate functions.

5. The biological relevance of the experiments in Figures 5 and 8 and respective supplemental figures is unclear to me: what have we learned of the biological functions of GNOM in development from those experiments? It seems to me that what is being reported is only observed in the presence of BFA; how do the observations translate into how the GNOM protein functions in normal development, i.e. in the absence of BFA?

– ll. 55-57. "These resemble some of the phenotypes caused by the disruption of polar auxin transport or auxin homeostasis, including the phenotypes of pin mutants deficient for PIN auxin efflux carriers (reviewed in Adamowski and Friml, 2015)." I am not sure that is true: the only thing pin mutants have in common with strong gnom mutant alleles is that they are small, have fused cotyledons, and are seedling lethal. But so are many other mutants. And in contrast to strong gnom mutants, pin mutants have a hypocotyl and an abnormal root – both of these structures are replaced in strong gnom mutants by a basal peg. As for the vascular defects of gnom and pin mutants, they are quite distinct. Please modify your statement by being more specific: what phenotypes are you referring to?

– l. 79. "To serve this function". In service of this function? To perform this function?

– Figure 1A. I appreciate that the fraction at the bottom right of the pictures represents reproducibility of plasma-membrane localization; however, it does look weird to see an image whose reproducibility is "0/57".

– ll. 207-212 refer to a difference between the phenotype of GNpro:GNfwr-GFP;gnS and gn-fwr but that difference is not shown; please do so.

– ll. 207-212. The argument hinges on the hypothesis that the level of the gn-fwr protein is higher than in the original gn-fwr mutant. This should be shown or the claim should be more circumspect.

– ll. 207-212. The argument is only valid for a subset of GNOM functions.

– In several instances of the Results sections, the authors refer to manuscripts in preparation or submitted instead of presenting the evidence, which is unacceptable because reviewers are unable to independently evaluate those claims.

– ll. 519 and 520. "This phenotype may be caused by a slight defect in polar auxin transport". The claim is entirely unsupported and speculative: please remove, modify, or support with evidence.

– ll. 651-653. It's unclear to me which findings in this manuscript "argue against a model where GN promotes an ARF-dependent formation of exocytic vesicles trafficking PINs from an intracellular compartment to the polar domain at the PM". Please elaborate.

[Editors' note: further revisions were suggested prior to acceptance, as described below.]

Thank you for resubmitting your work entitled "Developmental patterning function of GNOM ARF-GEF mediated from the plasma membrane" for further consideration by *eLife*. Your revised article has been evaluated by Jürgen Kleine-Vehn (Senior Editor) and a Reviewing Editor.

The manuscript has been improved but there are some remaining issues that need to be addressed, as outlined below:

All three reviewers find that the revised manuscript has been substantially improved. However, in our consultation session, they raised the issue that these results dramatically redefine the well-established localization of GNOM at the Golgi and the established function of GNOM in endocytosis. There have already been at least three studies of GN localization and its biological relevance (Geldner 2003, Naramoto 2010, and Naramoto 2014) and each of those studies corrects the previous one, and these new results contradict these other publications. As such, the reviewers agreed that two essential revisions are required:

1) It is essential to document whether GN and gn-fwr are in the PM or in cortical vesicles just below the PM using confocal z-stacks and 3D colocalization with a homogeneously distributed PM marker (i.e. something like PIP2A or LTI6b, rather than a heterogeneously distributed marker like clathrin components). Reviewer 1 has provided clear details about what is required here.

2) It is also essential to provide clear details about the GN-GFP line and articulate what is different about this line, compared to previously published GN-GFP lines. Reviewer 3 has provided clear details about what is required here.

*Reviewer #1 (Recommendations for the authors):*

In this revised version of the manuscript, the authors have added significant supplemental data addressed some of the points that I raised in my previous review. Importantly, they have added analysis of root hair positioning and of GNfer-GFP localization in trichoblasts, which clarifies that the GNfer-GFP construct can rescue even subtle defects of gn mutants. The manuscript text has been revised and several points have been clarified with these text revisions.

My main concern remains that the authors have not adequately demonstrated the localization of the GN puncta. They have provided no confocal stacks of GN or GNfer and they have not conducted or 3D colocalization with PM markers. So, it remains unclear whether (1) these GN puncta are in the PM or subcortical and (2) whether there really is no detectable signal in the cytosol for GNfer. These data are essential to support their claims of GN activity at the PM. They must either present data to address this question or dramatically revise the manuscript to acknowledge these caveats.

For point 1, in their response to reviewers, they say there is no reason to expect colocalization with any PM marker, but I believe they mean this in 2D, in the plane of the PM, and are not considering the question of whether the GN puncta may be subcortical, which could be resolved with 3D colocalization. They also say that they "are experienced in using TIRF microscopy and are confident that we observe cell surface-associated GN signal" but personal experience is insufficient to support such an essential claim. They contrast the size and speed of GN puncta to Golgi or endosomes, which are obviously much larger, much deeper in the cell, and moving quickly. My question is whether the GN puncta might be subcortical vesicles, which would still be small and visible in TIRF (e.g. late-stage CCVs, which they can clearly see in their CLC imaging in Figure 3A).

For point 2, they must present 3D confocal stacks of GNfer-GFP. I appreciate that they won't be able to observe the PM heterogeneity of GNfer-GFP with this technique compared to their very nice TIRF data, but these are essential data to support the claim that it GNfer-GFP "localized solely to the PM" (e.g. line 164), which is a critical point for this manuscript.

They have also failed to provide evidence of three independent transgenic lines for each construct, which is usually the absolute minimum. This is particularly important to support the conclusion that GNfer-GFP is "functional", which is also central to this manuscript.

Finally, images in main figures seem to have been re-cropped and reused in supplemental figures in several cases (for example, but not limited to, Figure 1 – Supplement 1A and Figure 2A; Figure 8A and Figure 8 – Supplement 1A). Although these figures describe the same information, I would hope that the authors have more than one image of these and can replace the reused images with unique data.

*Reviewer #2 (Recommendations for the authors):*

The revised version of the manuscript has greatly improved the logic and elaboration of BFA-triggered GN-association with the exocytosis pathway, in particular as a way to distinguish GN and GNL1 in their localization and functional regulation. The responses to the previous concerns were accepted to be reasonable for this reviewer. Although the story does not necessarily give an exciting mechanistic understanding about how GN functions in development, the establishment of its function domain at the subcellular level (plasma membrane) is an important contribution.

Lines 112-113 or 132-133, about the newly generated GN/GNL-GFP lines, more explanations are needed. Since the new localization of GN based on this marker like was the main discovery of the story, it is essential in the main text to explain whether and how this transgene is different or advanced from the previously established GNpro:GN-GFP line that has been widely used in the community and well demonstrated for main association with the GA (Naramoto et al., 2010).

Figure 1B, two panels need label for "weakly fluorescent, small and dynamic structures, probably residing in the cytosol" (Line 140) for both GN-GFP and GNL1-GFP.

Lines 150-151, why VAN3 was compared with GN-GFP should be explained in the main text.

*Reviewer #3 (Recommendations for the authors):*

Dear Jiri, Dear Maciek, Dear Ivana (I hope you will not mind if I address you by your name: after all, I know who the authors of the manuscript are, and I have agreed to disclose my name, so you know mine),

I am quite satisfied with how you addressed my concerns, and I only have three points left.

1. I think you should provide detailed information on how you constructed the GN-GFP line you used throughout this study. For example, you should provide the coordinates of the upstream sequence used, those of the gene or cDNA sequence used, those of the downstream sequence, the precise location of the GFP insertion, the presence of linkers, etc. Alternatively, you could provide the entire sequence of the GN-GFP transgene. This piece of information is essential to this manuscript because so much of what you conclude depends on that construct, and the information you have now provided in Supplementary File 1 is insufficient for researchers to independently replicate your findings.

2. On ll. 205-220, you wrote: "In our view, even if, following the fwr phenotype, GNfwr is considered as an incompletely effective GN variant, unable of promoting normal LR formation or correct vascular patterning, but functional in embryonic and most of post-embryonic development, the presented argument that PM, rather than the GA, is the site of GN action in developmental patterning, remains valid."

I maintain, as I did in my review to the originally submitted manuscript, that the argument is valid only for a subset of the patterning functions of GN, not for all known functions of it. For example, we do not know what the localization is of gn-fwr during vein or lateral-root patterning. Let's suppose we perform that experiment and find that gn-fwr is not localized to the PM during vein or lateral-root patterning. That finding would be consistent with your conclusion that the PM and not the GA is the action of GN's biological function. However, let's suppose that we find that gn-fwr is localized to the PM also during vein or lateral-root patterning. In that case – because gn-fwr has defects in both vein and lateral-root patterning – we would no longer be able to infer that PM-localization is sufficient for all patterning functions of GN. Of course, we would also be unable to infer that PM-localization is not sufficient for those functions of GN since the reduced funcionality of gn-fwr may be due to reasons other than its subcellular localization. Either way, however, the logic of the argument you propose would no longer be sufficiently stringent, and the argument would cease to be necessarily valid. Therefore – I ask – why not be circumspect and propose that it is possible that other functions of GN, untested in this study, may have different localization requirements?

3. Because fwr is a mutant allele of GN, shouldn't it be referred to as gn-fwr and not as GN-fwr throughout the manuscript and figures?

---

## [Author Response]

Essential revisions:1) To assess functionality of GN variants, the authors used the rescue of the seedling phenotype, including its lethality, of a strong gn mutant allele; however, those phenotypes only reflect near-complete loss of GN function. Indeed, different fluorescently tagged GNOM protein fusions have been generated in this manuscript and others, and all the fusions have been claimed to be fully functional based on their ability to rescue the seedling phenotype of a strong gnom mutant allele. Therefore, all the fusions seem to share "full functionality"; however, they also have different localization patterns. The most sensitive indicators of even partial loss of GNOM function are lateral root formation (Geldner 2004; Okumura 2013), root hair polar positioning (Fischer 2006) and vein network formation (Geldner 2004; Verna 2019); therefore, analysis of at least one of those diagnostic criteria should be included. At this moment, claims of full functionality of GNOM variants are not fully supported and prevent deep understanding: even gnom-fwr and gnom-B/E mutants look normal at the level that the authors have analyzed the functionality of GNOM variants in this manuscript.

We analyzed root hair positioning in *gn^s^* mutants rescued with GN-GFP, GN^fwr^-GFP and additionally the chimera LGG-GFP, which based on previous experiments we considered fully functional. All lines present normal root hair positioning near basal ends of epidermal cells, in contrast to the previously reported apical relocation in partial loss-of-function alleles of *gn*. The results are included in Figure 1—figure supplement 1.

2) Throughout the manuscript, localization of fluorescently tagged GNOM protein was determined in epidermal cells of either hypocotyl or roots. In these cells, GNOM function has never been shown to be required. Therefore, the biological relevance of GNOM localization in the imaged in this manuscript is unclear. This point could be addressed by:2a. GN-GFP localization in the cells in which GNOM is known to function, i.e. those cells that participate in the processes that most suffer from loss of GNOM function.2b. Document that indeed GNOM has functions in the cells that were used to determine GNOM localization and use such functions as diagnostic criteria of phenotypic rescue.2c. Some combination of 2a and 2b; for example, examining GN-GFP localization during root hair positioning (Fischer et al. 2006 Curr Biol) and carefully quantifying the ability of different GN-GFP constructs used in this paper to complement this mild gn phenotype.

Addressing point 2a was not feasible due to technical limitations: GN localization to the PM in the epidermis, and the absence of GN^FWR^ from Golgi structures, was in our experience detected most reliably with TIRF microscopy, and this cannot be evaluated in tissues other than epidermis. Therefore, as suggested in point 2c, we focused on testing the functionality of GN-GFP fusions in the epidermis during root hair positioning. As discussed in response to point 1 above, GN-GFP and GNfwr-GFP fusions were functional in this phenotypic aspect. In two independent lines of GN-GFP *gn* and GNfwr-GFP *gn,* we conducted additional TIRF imaging specifically at basal ends of young trichoblasts just before root hair outgrowth, i.e., where GN presumably acts in the positioning process. This showed a picture similar to previous imaging, i.e. both GN-GFP and GNfwr-GFP were observed at PM-localized structures. While the mechanism of action of GN required for root hair positioning is not known, the data is consistent with the scenario that also in this process, GN acts from the PM. The new data are now presented in Figure 2—figure supplement 1.

3) Steady-state localization of GNfwr-GFP at PM doesn't mean it's never intracellular (absence of evidence is not evidence of absence). This claim should be supported by quantitative analysis of multiple confocal stacks of whole cells (from multiple independent transgenic lines) to document whether any intracellular signal is detected and/or colocalization of PM puncta with a PM marker (e.g. PIP2A, LTI6b, WAVE131, WAVE138, or very short (<5min) FM4-64 treatment).

The proposed experiments rely on CLSM, while the most reliable information about GN and GNfwr sites of action was gained with TIRF microscopy since this method clearly detected Golgi-associated signals of wild type GN, as well as GNL1. The conclusion about the absence of GNfwr-GFP at the Golgi apparatus was based on a side by side comparisons with GN-GFP. Conditions of illumination were the same in each case, and GN-GFP was always seen at the Golgi apparatus, while GNfwr was never observed at any intracellular structures, either reminiscent of Golgi apparatus or otherwise, and the only intelligible pattern of localization were the PM-localized objects, smaller than Golgi apparatus and not mobile like organelles in the cytosol. Therefore, due to the use of GN-GFP control, the absence of evidence for GNfwr-GFP localization at the Golgi does constitute evidence that GNfwr-GFP is absent from this organelle. This comparative imaging was conducted in roots and in hypocotyls with identical conclusions. Additional trials with very high laser and high detection settings were conducted to detect any possible weak signals of GNfwr at the Golgi apparatus, which also did not lead to detection.

If, as Reviewer 1 suggests, GNfwr localizes to Golgi transiently, while only its steady-state localization is at the PM, the conclusion that GNfwr activity is not mediated from the Golgi apparatus stands, since there is no reason to associate the function with a minor, and, as discussed, never detected, site of localization.

Regarding the proposed colocalization of PM puncta with PM markers or short FM4-64 treatments, there is no reason to expect specific co-localization of GN-positive structures with FM4-64, as the dye stain all membrane non selectively. Similarly, there is no justified reason to expect that the objects colocalize at the PM with the proposed proteins. Our analysis of the PM localization pattern of GN with TIRF microscopy extends beyond the level of resolution where any PM-localized objects colocalize, as it appears when proteins are imaged at lower resolution with CLSM. That said, we have already colocalized GN with a PM marker – CLC2 – and their lack of exact colocalization well illustrates this point.

We have additionally addressed these issues in response to Reviewer 2’s issue 2.

4) Both Reviewers 1 and 2 have requested further quantification of imaging experiments from multiple (preferably 3 or more) independent transgenic lines – please see their comments details on which experiments require further quantification and statistical analyses.

This is addressed in individual cases, below.

5) Both Reviewer 1 and 3 raised questions about the experiments presented in Figures 5 and 8. Please revise the rationale, results, and conclusions around these data to clarify the biological questions being addressed by these experiments.

As admitted in the manuscript, the phenomenon of BFA-induced exocytosis of GNOM does not, in our view, directly inform on the native mode of action of GN. As explained in lines 411-415, the value of this BFA-induced observation is such that it indicates a physical association of GN, compared with GNL1, with particular components of the endomembrane system with which in likely traffics on secretory vesicles towards the PM. Considering that these association is specific to GN but not GNL1, and promoted by its domains as Figure 8 shows, they may be indicative of interactions with unknown molecular players that function in GN-mediated patterning. These components could be a subject of future studies based on protein-protein interaction or proximity labelling approaches, performed comparatively between GN and GNL1. We explained this better in the revised version of the manuscript and made efforts to make this part of the text more concise.

Reviewer #2 (Recommendations for the authors):In this paper, Adamowski and Friml present and investigate the observation that GNOM (GN), an ARF-GEF presumed act at the Golgi apparatus and/or endosomes, is localized to both intracellular structures (i.e. Golgi apparatus/TGN/endosomes) and the plasma membrane (PM). Although the observation that GN is partially localized to the PM is not new (Naramoto et al., 2014 Plant Cell), the authors describe this localization in more detail and hypothesize about what role GN might play at the PM. Importantly, they find that a GFP fusion to the protein encoded by a weak mutant allele of GN localizes primarily to the PM and can complement most gn knockout phenotypes, implying that most of GN's role is performed at the PM (or, at least, that other intracellular ARF-GEFs act redundantly with intracellular GN, but not PM GN). Domain-swap experiments between GN and GNL1 document that no single region/domain is responsible for the functional or localization differences between GN or GNL1. Their claims are supported by molecular complementation of different mutant backgrounds (usually with two independent transgenic lines) and by high quality live cell imaging experiments. Most of the presented evidence are qualitative observations and some experiments are appropriately quantified.The authors assume that the unique molecular function of GN is related to this PM localization, and that any intracellular functions performed by GN are redundant with GNL1. This hypothesis separates GN function from a direct influence on PIN trafficking from the endomembrane system, and rather implicates GN in the establishment/maintenance of some other unknown polarity cue that will subsequently affect PIN trafficking/localization. As a result, these findings potentially have important implications for the study of auxin, and therefore plant growth and development. Clearly, many questions remain about what polarity cues GN might influence and the mechanisms by which GN exerts its influence at the PM.Comments to Authors:Overall, this manuscript documents and investigates an interesting observation that GN is partially localized to the PM. These results have important implications for plant biologists studying auxin, intracellular trafficking, and plant cell polarity. Below, I have raised several points that the authors should address to improve the connection between author claims and evidence (items 1-4, 6), to provide appropriate quantification (item 5), to clarify the methods (items 7 and 8).1. Steady-state localization of GNfewerroots-GFP at PM doesn't mean it's never in the Golgi/TGN (absence of evidence is not evidence of absence). At the very least, this claim should be supported by quantitative analysis of multiple confocal stacks of whole cells (from multiple independent transgenic lines) to document whether any intracellular signal is detected.

We responded to this in Essential revisions, point 3, above.

2. I am confused by the notion of GN being "specifically exocytosed" (line 19) "anterograde secretory movement of GN through the endomembrane system" (line 349), "GN-exocytosis" and the "BFA-induced exocytic process"; isn't GN a soluble protein? How is it "in" the endomembrane system? Is it not just peripherally associating with membranes? I think the text explaining the rationale, results, and conclusions around figures 5, 8, and S3 need to be substantially reworded to clarify this. I think it's very important to clarify whether GN localizes to the PM via vesicle trafficking (i.e. "secretion" or "exocytosis") or via peripheral association with PM proteins (which may be the thing that is actually mislocalized during BFA treatment of big3 mutants).

Naturally, GN is a cytosolic protein, which associates with membranes peripherally. When referring to BFA-induced exocytosis of GN, we mean a process where GN secretes while bound peripherally to membrane structures, through interactions with membranes and/or with proteins present on, or in, the membranes of the Golgi, the TGN, secretory vesicles, and finally the PM. Thus, we envision GN to traffic on the surface of exocytic vesicles to reach the PM in these conditions. We now clarified this better in the text.

We recognize that it is theoretically within possibility that only the proteins with which GN associates become secreted to the PM, while GN binds to these proteins directly from the cytosol at the PM directly from the cytosol. This proposition is, however, less likely than the scenario proposed by us, seeing how the GN association with all membranes, also at intracellular compartments, increase when this ARF-GEF is inhibited by BFA (e.g. Figure 5A).

3. It is unclear why the authors chose to use the GNpro:GN-GFP VAN3pro:VAN3-mRFP line for later experiments (Figure 5), particularly since it shows far less association with the PM (Figure 5A vs Figure 1A) the GNpro:GN-GFP line. Please clarify.

The use of the new GN-GFP fusions would be here, naturally, optimal but the difference between the GN reporter lines was realized only late in the course of the project. Crosses with the already available GNpro:GN-GFP VAN3pro:VAN3-mRFP line, used in Figure 5 to study the phenomenon of BFA-induced secretion, were generated at a point before the new GN-GFP lines were scrutinized in detail. The current use of lines does not have any imaginable impact on the conclusions drawn.

4. How many unique transformation lines were observed to draw the conclusion that GN-GFP is "always" partly localized to the PM, and how many GNL1-GFP lines were observed to support the conclusion that GNL1 is "never" found at the PM? I see evidence of only two GN-GFP lines in Figure S1, but it is unclear whether more than one line was used for imaging. Similarly, for the chimeric complementation experiments, the authors state that they analyzed two independent transgenic lines; three is usually the absolute minimum number, but even so, the data presented in Figure 6-8 seem to be from only one line. Please provide evaluations of multiple independent transgenic lines in supplemental data.

Both CLSM and TIRF comparisons of subcellular localizations between GN-GFP and GNL1-GFP were performed on two transgenic lines expressing each protein fusion, with identical conclusion. The results presented in the Figure illustrates an observation common to both lines analysed, and quantitative data a sum of all observations. This information was added to manuscript text and to figure legend. Additional images from each individual transgenic line were shown in a figure supplement.

With regards to chimeras, Figure 6 shows adult plants from two transgenic lines, which was now clarified in the Figure legend. Second line of GLL-GFP was added to the figure. Data from two transgenic lines for seedling phenotypes in Figure 7 was now shown in supplement. Imaging data from the transgenic line used in Figure 8 was shown in supplement. Quantification in Figure 8C is a sum of two transgenic lines tested as was now indicated in Figure legend.

5. Many of the claims are qualitative and could easily be supported by quantitative data:­ That GN localizes to "often stable" punctae in the PM – what is the average (and range) of the size of these particles, their density in the PM, their lifetime, their speed of movement, and their range of movement?

Particle sizes were not measured due to limitations of fluorescent microscopy. Close inspection revealed that the apparent size differs depending on fluorescence detection level in individual pictures, in that higher exposed samples appear to present larger objects. In any case, given their relatively miniscule size compared with image resolution, a measurement would likely be far from accurate. The description of particle sizes will have to be performed when their physical nature is defined and their existence identified by electron microscopy.

The densities of GN-GFP and GNfwr-GFP particles were measured. It was found that GNfwr-GFP shows a higher average density.

At this point, life times could not be precisely measured as the vast majority of objects remain at the PM for time periods exceeding the total time of imaging. Please, see kymographs in the manuscript for examples showing this. To provide at least an indicative life time information, we now stated in the text that lifetimes above 100 seconds were common.

The range of lateral movement was typically very small relative to pixel resolution of the images. It was nevertheless estimated by the measurement of lateral displacement of tracks on kymograph maximum projections. The values were the same for GN-GFP and FWR-GFP.

Total lateral displacement was typically small relative to the resolution of captured movies, and movements tended to oscillate, as seen on presented kymographs. For these reasons the measurement of movement speed was not feasible as the values would be close to zero and accuracy of measurement very limited.

We understand and strongly identify with the demand of supporting any claims by quantitative data whenever possible and feasible. Nonetheless, this request should not be done without considering the context of the situation. In the revised version, we included quantitative information wherever appropriate and we are very confident that all claims related to the localizations are well supported by observations.

­– That GN signal is enhanced at the PM after BFA treatment (line 163): what is the change in mean signal intensity? How are these particles different from those observed under control conditions: what is the average (and range) of the size of these particles, their density in the PM, their lifetime, their speed of movement, and their range of movement?

The increase in PM signal intensity of GN-GFP after a BFA treatment is shown on CLSM pictures in Figure 5A with measurement in Figure 5E. Another set of repetitions with further measurements is shown in Figure 8C. We realize our mistake of not referring to the signal measurement in Figure 5E at this point of the text. This was now corrected. A measurement of signal intensities in TIRF microscopy is not possible due to the fact that illumination settings (the laser incidence angle and the resulting level of fluorescent protein excitation) are adjusted individually to each imaged root or hypocotyl sample.

Similarly to the case discussed above, the size of these particles could not be reliably measured with the present methods. Another difficulty is the very high density of these objects and a higher background signal in BFA-treated samples.

The density of these particles was measured against an untreated GN-GFP control and was found to be higher.

Regarding their lifetime and movements, as described in the manuscript text, as well as shown in the kymograph on Figure 1D and the supplemented movie, the particles present after a BFA treatment behave very dynamically, i.e. appear and disappear rapidly. Their appearance is often limited to single frames of movies, or, in other words, single pixels in the vertical direction on kymographs. While no measurement can be obtained from such data, the qualitative difference from untreated conditions is very obvious and is described in the manuscript and, we hope, is sufficient to understand the observed phenomenon.

– That gn knockouts expressing GNfwr are indistinguishable from gn knockouts expressing GN-GFP, which are indistinguishable from wild type: please quantify root length and test for statistically significant differences between multiple independent transformants and wild type.

Root lengths were quantified and no reduction in root lengths was found in *gn* complemented by GNpro:GN-GFP and GNpro:GNfwr-GFP, compared with Col-0. However, One-way ANOVA with post-hoc Tukey’s HSD test revealed an increased root length in GN-GFP L2 compared with Col-0 and GN-GFP L1. This, however, is not indicative of an incomplete GN-GFP transgene function in this phenotypic aspect.

– That GNfwr-GFP behaves in a manner indistinguishable from GN-GFP at the PM – what is the average (and range) of the size of these particles, their density in the PM, their lifetime, their speed of movement, and their range of movement? Are there any statistically significant differences between these metrics for GNfwr-GFP compared to GN-GFP?

Please see above.

– That GN-GFP is not colocalized with CME machinery (line 238): please quantify colocalization, and include an appropriate control (e.g. colocalization of 2 CME components)

Co-localization of structures positive for fluorescent proteins captured in TIRF movies is purposefully described in qualitative terms. Using time lapse movies and by kymographs, we describe the character of the structures in space and time, and observe that GN-positive structures are simply distinct entities from clathrin-coated pits. Even if some pixel colocalization were to be measured in single TIRF frames, this would not affect the conclusion.

To strengthen this point, we performed a control co-localization of 2 CME components to show qualitatively how fluorescent markers associated with clathrin-coated pits present themselves in TIRF imaging. An evident co-localization in space and time was observed.

– That bulk secretion is unaffected by loss of GN (line 274): please use a quantitative measure of secretion, for example, ratiometric secGFP (Samalova et al. 2015, Traffic) and quantify using both fluorescence ratios and Western blots. There seems to be a lot of signal aggregates in the gn seedling, which would actually suggest that secretion is affected. Please also include a wild type control at a similar developmental stage.

We included additional controls to further validate our conclusions. We analyzed secRFP in RAMs of 2 d old wild type seedlings which are more closely comparable with gn knockouts. We ruled out the possibility that the observed intracellular signals are aggregations of secRFP by a control imaging of *gn* not expressing secRFP. These structures are likely plastid autofluorescence which we were unable to separate from secRFP signals.

If the Reviewer and the Editors consider these efforts insufficient to support our statement that secretion is grossly normal in *gn* knockouts, we propose to remove this section of results, as this would have no strong impact on the rest of the manuscript, and was rather provided for completeness of GN analysis as a component of the endomembrane system.

– The histogram in Figure 3D would be more appropriate as a violin plot. Please also use an appropriate statistical test to determine whether "event lifetime" is "the same as in wild type controls" (line 266) for both CME markers.

We present event lifetime data as histograms following classics (Konopka et al., 2008, The Plant Cell, Vol. 20: 1363–1380).

To clarify, we do not claim that event lifetimes are the same as wild type for both CME markers. We state this only in relation to TPLATE-GFP, but not CLC2-GFP. However, we do not believe that supporting our observations with statistical tests would be additionally informative. The *gn* mutant is characterized by a strikingly strong phenotype. If the *gn* defect were caused by a dysfunctional CME, then CME would be expected to be defective in a similarly major degree. Our evaluation of CME in the mutant was intended to address this eventuality, and the finding was opposite: It is remarkable how normal CME appears in these hardly developing mutant seedlings. Minor differences, such as were observed with CLC2-GFP, and might be detected by statistics with TPLATE-GFP, would not change our conclusion that GN developmental function is not strongly tied with CME. We explain this reasoning in the revised version.

6. References to Adamowski and Friml 2021a (submitted), Adamowski and Friml 2021b (submitted), Adamowski and Friml 2021c (in preparation), and Adamowski et al., 2021 (in preparation) must be removed or replaced with references to published articles or published preprints. Any claims supported by these citations should be supported by references to published articles or published preprints; otherwise, data should be presented in this manuscript to support the claims, or the claims should be removed from the manuscript.

Any reference to two of these manuscripts was removed from the manuscript, while two other manuscripts were published as preprints.

7. Many essential details are missing from the methods section and must be added:– Please provide catalogue numbers for critical chemicals (e.g. BFA, β-estradiol).– Light microscopy: details of the objective and camera used for imaging seedlings.– CLSM: NA for the objectives used, excitation and emission wavelengths, z-spacing for any z-stacks, details on the detector type and detector settings used for image acquisition (e.g. PMT type with info on gain, offset, scan frequency, any line/frame averaging performed during collection).– TIRFM: NA of objective, angle of illumination, excitation and emission wavelengths, details on the detector type and detector settings used for image acquisition (e.g. EMCCD or sCMOS type and settings).

These details were provided now.

8. Finally, please check the name of the gn T-DNA allele on line 136; Salk_103114 directed me here (https://abrc.osu.edu/stocks/number/SALK_103114) which is a T-DNA line affecting CESA2 and/or pre-tRNA.

Thank you for pointing out this error. The line is SALK_103014.

Reviewer #2 (Recommendations for the authors):The manuscript entitled "Developmental patterning function of GNOM (GN) ARF-GEF mediated from the plasma membrane" by Adamowski and Friml elaborated the analyses of GNOM localization at the plasma membrane. By using the TIRF microscopy, GN was determined to accumulate to unknown, relative static puncta at the plasma membrane. The authors further showed that the CME and bulk secretion processes appeared normally in gn mutants. Additionally, through domain exchange experiments, chimeric ARF-GEFs of GN-GNL1 were evaluated for their biological functions in vivo, leading to the conclusion that all domains of GN contribute to GN function.The ARF-GEF GNOM has been known for its predominant localization at the Golgi apparatus to regulate secretion/recycling, thereby promote the polarization of PIN proteins at the plasma membrane in plant cells. Although a few previous publications reported the presence and function of GNOM at the plasma membrane (PM) (Naramoto 2010 and Okumura 2013), this study provides more in-depth cell biological analyses and established the localization pattern and dynamics of GN at the PM. These discoveries bring new insight into how GN may promote PIN polarization at differential subcellular locations, in particular the underexplored contribution from the plasma membrane pool. I am convinced that GN is partially localized to the plasma membrane beside the previously established localization at the Golgi but have the concern that the claim of "PM is the major place of GN action responsible for its developmental function" might be an overstatement.1. The main basis for the authors' claim was (1) GN-GFP is partially localized to the PM as described in this story and previously in Naramoto 2010. The enrichment of GN-GFP at the PM was elevated by BFA treatment. (2) the localization of a GN mutant variant, GNfwr, is predominant at the plasma membrane. GNfwr contains a point mutation in one of the HDS domains that was screened for the mutant phenotype in lateral root initiation and abnormal patterning of auxin signaling in root development. Although GNfwr can largely complement the growth phenotypes of loss-of-function gn mutants, I doubt whether this provides the most solid evidence to make the conclusion. In any case, it remains unclear about the nature of the GNfwr defects and how GNfwr becomes more PM-localized. Was this due to interfered protein-protein or protein-membrane interactions, or due to the abnormal membrane trafficking when GNfwr is not able to localize to the Golgi? To make the experimental data more convincing, I would create situations that the wild-type GN protein can be functionally tested at the PM solely (without Golgi association) or at the Golgi solely (without PM association) in gn mutants. The localization and function of GNfwr might be too complicated to be heavily based on.

Next to the direct comparison between localizations of GN and GNL1, the GNfwr variant remains the strongest argument for the PM site of action of GN. At this point, the cause of GNfwr localization is not clear. Understanding this mechanistically would be an important step in understanding the activity carried out by GN. It is a good question whether the exclusive PM localization is caused by distinct molecular interactions, but interactors of GN are virtually unknown at this point. While the reason behind GNfwr localization is not clear, our argumentation is logical: a GN variant cannot be detected at the Golgi apparatus, but only at PM structures specific to GN and not GNL1, and that GN variant rescues the *gn* mutant to a very high degree. Modulating wild type GN localization is a very interesting suggestion, but we do not see a way to do this at this point. Therefore, we put all effort in detailed characterization of GNfwr to be able to make the interpretation with confidence.

2. I wonder whether there is a way to better present the punctate structures that are indeed localized to or tightly associated with the plasma membrane but not underneath the membrane and in the cytoplasm (the TIRF images in Figure 1). Do these dots co-localize with the FM4-64 dye after a short-time treatment, or do they overlap with any of those Wave endosomal markers?

We are experienced in using TIRF microscopy and are confident that we observe cell surface-associated GN signal for several reasons. With TIRF, we can set the microscope so that only signals at the cell surface are well-illuminated. The structures below are partly visible only if there are large and bright, and in such case, they move along with cytosolic streaming. Studies of CME machinery by TIRF microscopy show how PM-localized proteins tend to appear as structures that do not rapidly move laterally through captured movies, in contrast to intracellular structures that flow with cytosolic movement. If GN-positive structures would not be attached to the PM, they would move along with other subcellular structures. Furthermore, endosomal compartments, for instance the early endosome stained with CLC2-GFP, or Golgi apparatus stained with GN-GFP, present a different appearance, as they are much larger, and, as discussed just above, mobile in the cytosol. In any case, the GN-positive structures are detected in the outer surface of the sample in TIRF movies, but not when the objective is focused deeper in the cells. Endosomal compartments, in contrast, show opposite behaviour.

Furthermore, co-localization with FM4-64 cannot be expected to be very meaningful, since this dye stains all membrane non-selectively, and would stain very quickly also early endosomes in the PM vicinity. We tried it as suggested and indeed the outcome was non-informative.

Please see our response to Essential revisions, point 3, for additional comments on this topic.

Reviewer #3 (Recommendations for the authors):The cellular localization of the GNOM protein of Arabidopsis and the biological relevance of such localization has been object of interest for the past 20+ years. During this time, our knowledge of where in the cell the GNOM protein is and acts has continually been refined. In this manuscript, Adamowski and Friml continue to address this question and now suggest that the main site of action of the GNOM protein is not any of the intracellular membrane compartments, as previously thought, but the plasma membrane.The gnom mutant is arguably the most severe patterning mutant in Arabidopsis – in the most extreme case, gnom mutant seedlings are nothing more than spheres seemingly lacking any polarity. As such, GNOM function is key to plant development, and understanding the cellular site of action of the GNOM protein is certainly one important step toward understanding such key function. Therefore, this study has the potential to advance and deepen our knowledge of how plants develop; however, for the following reasons the potential of the study is not yet fully realized.1. In this manuscript, functionality of GNOM variants was assessed by their ability to rescue the seedling phenotype of a strong gnom mutant allele, including its seedling lethality. However, the most sensitive indicators of even partial loss of GNOM function are lateral root formation (Geldner 2004; Okumura 2013) and vein network formation (Geldner 2004; Verna 2019); therefore, analysis of those two diagnostic criteria – with which the authors are familiar – should be included. At this moment, claims of full functionality of GNOM variants are not fully supported and prevent deep understanding: even gnom-fwr and gnom-B/E mutants look normal at the level with which the authors have analyzed the functionality of GNOM variants in this manuscript.

This is addressed in Essential revisions point 1.

2. Throughout the manuscript, localization of fluorescently tagged GNOM protein was determined in epidermal cells of either hypocotyl or roots. In those very epidermal cells GNOM function has never been shown to be required. Therefore, the biological relevance of GNOM localization in the epidermal cells of root or hypocotyl imaged in this manuscript is unclear. Localization should be determined in the cells in which GNOM is known to function, i.e. those cells that participate in the processes that most suffer from loss of GNOM function. I very well realize that because of technical limitations this is not an easy point to address, but in the absence of such evidence the biological relevance of the localization patterns shown in this manuscript is unclear at best. Alternatively, the authors could show that indeed GNOM has functions in the cells that were used to determine GNOM localization and use such functions as diagnostic criteria of phenotypic rescue (see point 1 above).

This is addressed in Essential revisions point 2.

3. Different fluorescently tagged GNOM protein fusions have been generated in this manuscript and in previous ones, and all the fusions have been claimed to be fully functional based on their ability to rescue the seedling phenotype of a strong gnom mutant allele, including its seedling lethality. Therefore, all the fusions seem to share full functionality; however, they also seem to have different localization patterns. It's now important to evaluate claims of full functionality by assessing the ability of all the fusions to rescue the phenotype, including the lateral root and vascular network phenotypes, of the same gnom mutant allele: if some of those fusions were not localized to the plasma membrane and yet rescued all the phenotypes of a strong gnom mutant allele, including the lateral root and vein network phenotypes, plasma-membrane-localization of GNOM would not be relevant. Alternatively, it's possible that different developmental functions depend – to varying degrees – on GNOM localization to different cellular compartments.

Following the consensus Essential revisions, we additionally analyzed root hair positioning of *gn* rescued with newly generated GN-GFP fusions, where normal phenotype was found. While we did not analyse the functionality of the formerly generated GN-GFP fusion, published in Naramoto et al., PNAS, this fusion also shows PM-localized structures via TIRF microscopy (Figure 1—figure supplement 3). The only difference is the apparent less consistent detection of PM signals with CLSM.

The argument for PM site of action relies to a larger degree on the difference in localizations of GN-GFP and GNL1-GFP – there is no reason why GNL1-GFP should be absent from the PM, if this localization is unconnected to GN function – and on GNfwr-GFP, which, as extensively analyzed and discussed in the course of present revisions, cannot be detected at the Golgi apparatus. We now further analyzed functionality of GNfwr-GFP on the sensitive phenotype of root hair positioning where normal phenotype was found.

The only alternative which we can conceive would be an action of GN from the cytosol – unusual for a protein with membrane-binding ability – or from the observed dynamic vesicle-like structures, present also with GNL1-GFP. Yet, the sum of arguments presented make the PM site of action most likely.

4. In Figure 6, the authors investigate the question what domains in the GNOM protein are responsible for the functions that are unique to this protein and not to the GNOM-LIKE1 protein. To address this question, the authors create protein chimeras between GNOM and GNOM-LIKE1, and assess their ability to rescue the seedling phenotype of a strong gnom mutant allele, including its seedling lethality. This is an exciting approach, but also a preliminary one: not only should the phenotypic rescue include more sensitive diagnostic criteria (see point 1 above), but the number of chimeras generated is preliminary – for example, the study lumps together all the three HDS domains, as if they may not have separate functions.

We recognize that putting together all HDS domains is a simplification, and a more elaborate analysis could be performed in the future. We acknowledged this limitation in the text.

Following Essential revisions point 1, we analysed root hair positioning – a very sensitive *gn* phenotype – in LGG, as the only chimera which based on our criteria, we interpreted to be fully functional. We found root hair positioning to be normal.

5. The biological relevance of the experiments in Figures 5 and 8 and respective supplemental figures is unclear to me: what have we learned of the biological functions of GNOM in development from those experiments? It seems to me that what is being reported is only observed in the presence of BFA; how do the observations translate into how the GNOM protein functions in normal development, i.e. in the absence of BFA?

We agree that this observation is only limited to an artificial situation of a BFA treatment. This was acknowledged in lines 404-414 of Results and 626-634 of Discussion in the previous version. We find this observation valuable as a complement of the presented study that focuses on the activity of GN within the endomembrane system. This artificially induced process indicates that GN associates with proteins present on exocytic vesicles upon BFA treatment, in a manner selective in comparison to GNL1; therefore, these molecular interactors may be associated with the molecular function of GN in development. The observation warrants future protein-protein interaction studies, or proximity labelling studies, comparing GN and GNL1, in order to elucidate the mechanism of GN activity. Our interpretations are now more clearly explained in the text.

– ll. 55-57. "These resemble some of the phenotypes caused by the disruption of polar auxin transport or auxin homeostasis, including the phenotypes of pin mutants deficient for PIN auxin efflux carriers (reviewed in Adamowski and Friml, 2015)." I am not sure that is true: the only thing pin mutants have in common with strong gnom mutant alleles is that they are small, have fused cotyledons, and are seedling lethal. But so are many other mutants. And in contrast to strong gnom mutants, pin mutants have a hypocotyl and an abnormal root – both of these structures are replaced in strong gnom mutants by a basal peg. As for the vascular defects of gnom and pin mutants, they are quite distinct. Please modify your statement by being more specific: what phenotypes are you referring to?

It is correct that phenotypes of *pin* and *gn* mutants are to a large degree distinct, and we recognize as well as appreciate research inputs of the Reviewer into characterizing the function of GN as broader than the control of PIN polarity and transport activity. The sentence was meant to provide a historical narrative into why research on GN focused on connections to PIN proteins and indeed may have been imprecise. The sentence was now removed.

– l. 79. "To serve this function". In service of this function? To perform this function?

Corrected.

– Figure 1A. I appreciate that the fraction at the bottom right of the pictures represents reproducibility of plasma-membrane localization; however, it does look weird to see an image whose reproducibility is "0/57".

As indicated in Figure legend, the meaning of this label was different, not an information of reproducibility of the observation shown in the figure, but of frequency of an observation being compared between samples, in this case, the frequency of PM localization. We realize that this is a different presentation than the one often utilized by other authors, but we believe it to also be logical.

– ll. 207-212 refer to a difference between the phenotype of GNpro:GNfwr-GFP;gnS and gn-fwr but that difference is not shown; please do so.

We refer to the difference on the basis of the following facts:

the isolation of gn-fwr in a forward genetic screen, an approach which proves that the mutant exhibits phenotypes,images, quantifications, and descriptions published in Okumura et al., 2013.the fact that GN:GNfwr-GFP;gn^s^ is indistinguishable from Col-0 and GN:GN-GFP;gn^s^ controls, now supported by a quantification of root lengths following a suggestion of Reviewer 1.

– ll. 207-212. The argument hinges on the hypothesis that the level of the gn-fwr protein is higher than in the original gn-fwr mutant. This should be shown or the claim should be more circumspect.

A different level of expression is proposed as a possible explanation for the discrepancy of phenotypes, using the word “might” to indicate that this is only an untested possibility. We now also provided a second possible explanation, referring to altered expression patterns from the transgene. We are open to rephrasing this sentence further.

– ll. 207-212. The argument is only valid for a subset of GNOM functions.

We do not exactly follow this point, but will be happy to further adapt this argument to the suggestions of the Reviewer.

– In several instances of the Results sections, the authors refer to manuscripts in preparation or submitted instead of presenting the evidence, which is unacceptable because reviewers are unable to independently evaluate those claims.

This is addressed under Reviewer 1, point 6.

– ll. 519 and 520. "This phenotype may be caused by a slight defect in polar auxin transport". The claim is entirely unsupported and speculative: please remove, modify, or support with evidence.

The missing evidence was now presented.

– ll. 651-653. It's unclear to me which findings in this manuscript "argue against a model where GN promotes an ARF-dependent formation of exocytic vesicles trafficking PINs from an intracellular compartment to the polar domain at the PM". Please elaborate.

The manuscript argues that the subset of GN function which is needed for patterned development, which includes PIN polarity determination, is mediated from the PM, but not from the Golgi apparatus. If GN acts from the PM, then GN cannot directly promote formation of vesicles that traffic PIN from intracellular compartments to polar domains at the PM. The control of PIN polarity by GN must be indirect, for instance, may include the definition of polar domains, to which PIN is delivered by GN-independent vesicles. GN present at the Golgi apparatus together with GNL1 is likely to act in secretion, but this function is not essential for developmental patterning activity. We adapted the statement to make clear that it refers to its role in patterning.

[Editors’ note: what follows is the authors’ response to the second round of review.]

Thank you for resubmitting your work entitled "Developmental patterning function of GNOM ARF-GEF mediated from the plasma membrane" for further consideration by eLife. Your revised article has been evaluated by Jürgen Kleine-Vehn (Senior Editor) and a Reviewing Editor.The manuscript has been improved but there are some remaining issues that need to be addressed, as outlined below:All three reviewers find that the revised manuscript has been substantially improved. However, in our consultation session, they raised the issue that these results dramatically redefine the well-established localization of GNOM at the Golgi and the established function of GNOM in endocytosis. There have already been at least three studies of GN localization and its biological relevance (Geldner 2003, Naramoto 2010, and Naramoto 2014) and each of those studies corrects the previous one, and these new results contradict these other publications. As such, the reviewers agreed that two essential revisions are required:

Thank you for appreciating the improvements to our manuscript and pointing out the difficulties with determining the subcellular place of GNOM action during the last 20 years. However, we do not really feel that this one or the previous publications would contradict or invalidate the observations done before. If our manuscript made this impression, it is poor wording that we hopefully corrected in the new version. All the observations made in the previous publications still stand as they were done; it is just that both, our techniques and our understanding of subcellular trafficking mechanisms have evolved quite a bit in the last 20 years leading to a refinement of observations and a re-interpretation of the results. It is possible that the revision/evolution of our knowledge about GNOM place of action is more dramatic than for most other regulators but there are objective and understandable reasons for that. Brefeldin A induces translocation of GNOM from the Golgi to the TGN/endosomes, which the authors of Geldner 2003 were unaware of (as we were at that time) and this got explained or corrected (if reviewers wish so) by Naramoto 2014 with better lines and much better live microscopy. Naramoto 2010 made use of advance of TIRF microscopy happening at this time and did not correct but solely extend the Geldner 2003 showing additional place of action at the PM. This current study also does not contradict the previous ones but builds on them. It assigns to the GNOM pool at the GA the function in secretion – functionally redundant with GNL1, whereas the function in polarity and developmental patterning seems to rely on GNOM associated with the cell periphery. Thus, these findings are rather a functional extension of the previous studies, not their correction. In the revised version, we explained these points better and made our statements more in line with the previous findings.

1) It is essential to document whether GN and gn-fwr are in the PM or in cortical vesicles just below the PM using confocal z-stacks and 3D colocalization with a homogeneously distributed PM marker (i.e. something like PIP2A or LTI6b, rather than a heterogeneously distributed marker like clathrin components). Reviewer 1 has provided clear details about what is required here.

We elaborated on this point below.

2) It is also essential to provide clear details about the GN-GFP line and articulate what is different about this line, compared to previously published GN-GFP lines. Reviewer 3 has provided clear details about what is required here.

We elaborated on this point below.

Reviewer #1 (Recommendations for the authors):In this revised version of the manuscript, the authors have added significant supplemental data addressed some of the points that I raised in my previous review. Importantly, they have added analysis of root hair positioning and of GNfer-GFP localization in trichoblasts, which clarifies that the GNfer-GFP construct can rescue even subtle defects of gn mutants. The manuscript text has been revised and several points have been clarified with these text revisions.My main concern remains that the authors have not adequately demonstrated the localization of the GN puncta. They have provided no confocal stacks of GN or GNfer and they have not conducted or 3D colocalization with PM markers. So, it remains unclear whether (1) these GN puncta are in the PM or subcortical and (2) whether there really is no detectable signal in the cytosol for GNfer. These data are essential to support their claims of GN activity at the PM. They must either present data to address this question or dramatically revise the manuscript to acknowledge these caveats.For point 1, in their response to reviewers, they say there is no reason to expect colocalization with any PM marker, but I believe they mean this in 2D, in the plane of the PM, and are not considering the question of whether the GN puncta may be subcortical, which could be resolved with 3D colocalization. They also say that they "are experienced in using TIRF microscopy and are confident that we observe cell surface-associated GN signal" but personal experience is insufficient to support such an essential claim. They contrast the size and speed of GN puncta to Golgi or endosomes, which are obviously much larger, much deeper in the cell, and moving quickly. My question is whether the GN puncta might be subcortical vesicles, which would still be small and visible in TIRF (e.g. late-stage CCVs, which they can clearly see in their CLC imaging in Figure 3A).

Our TIRF setup indeed does not isolate the PM from the underlying cortical area. We did our best to address the Reviewer’s question as described below, but without success, despite lot of effort.

We tried to address this question also by using a super-resolution Zeiss Airyscan confocal microscope. We colocalized GN and GNfwr with FM4-64 membrane dye along with controls for PM (GFP-PIP2a) and cortical (MAP4-GFP, i.e. a microtubule marker) localizations. In this experiment, GN and GNfwr colocalized with the membrane dye. However, the cortical marker also overlapped with the membrane dye in many instances, which shows that a cortical localization cannot be reliably determined even with this super-resolution technique (we can provide the pictures demonstrating the above points, if needed). As such, we cannot consider the result regarding GN localization at the PM to be definitive.

As we recognize the Reviewer’s concern, we added both, in Results and Discussion statements regarding a possible cortical, rather than PM-bound, localization of these structures and throughout the manuscript we reformulated the GN localization as being at the cell periphery rather than definitively at the PM. We feel this doesn’t really affect the central message of our paper about two functions of GNOM: one in secretion being associated with GA and other, more mysterious and relevant to polarity and development, being associated with the cell periphery.

For point 2, they must present 3D confocal stacks of GNfer-GFP. I appreciate that they won't be able to observe the PM heterogeneity of GNfer-GFP with this technique compared to their very nice TIRF data, but these are essential data to support the claim that it GNfer-GFP "localized solely to the PM" (e.g. line 164), which is a critical point for this manuscript.

Our meaning was rather that GNfwr is localized to the cell periphery as opposed to the Golgi. Indeed, GNfwr does localize to the cytosol, in a form of a diffuse signal, as seen for instance in the confocal section shown in Figure 5C. This is similar to GN and GNL1 (Figure 1A, 1B), and is expected for ARF-GEFs as cytosolic proteins recruited to membranes by presumably transient protein-protein interactions. Our reasoning about the site of action of GNfwr is based on the fact that the cell periphery (most likely PM) foci are the only site where the protein clearly accumulates. Should GNfwr specifically bind elsewhere, inside the cell, this localization would be detected with our TIRF setup, just like Golgi signals of GNL1 and GN were reliably detected.

We corrected the text to additionally highlight the cytosolic presence of GN and GNfwr, and to remove imprecise statements such as that quoted by the Reviewer. We also added information about the generous amount of collected TIRF data that supports the conclusion about GNfwr localization. The numbers do not include additional trials with live preview only, where a very high laser was used to try and detect even trace signals on Golgi or other possible organelles.

They have also failed to provide evidence of three independent transgenic lines for each construct, which is usually the absolute minimum. This is particularly important to support the conclusion that GNfer-GFP is "functional", which is also central to this manuscript.

We initially isolated more independent lines and observed consistent phenotypes, such as a lack of complementation, a complete complementation at the adult stage, or consistent partially deficient phenotypes such as the long stems with relatively few organs of GLG, or the dwarf phenotype of LGL. As such, we have confidence that the presented very detailed characterizations of two lines selected out of those, are representative. With regard to GNfwr-GFP functionality, the fact that even a very sensitive phenotype of root hair positioning is complemented, is to us particularly reassuring. In the new version of the manuscript we recognized more clearly the limitations of using *gn* GNfwr-GFP lines and in the Methods section we described the selection process leading to the 2 presented lines from more independent lines showing the same overall complementation or non-complementation effect.

Finally, images in main figures seem to have been re-cropped and reused in supplemental figures in several cases (for example, but not limited to, Figure 1 – Supplement 1A and Figure 2A; Figure 8A and Figure 8 – Supplement 1A). Although these figures describe the same information, I would hope that the authors have more than one image of these and can replace the reused images with unique data.

Indeed, much more data were collected. We searched for all mentioned instances of image recycling and replaced the repeated images with a unique data.

Reviewer #2 (Recommendations for the authors):The revised version of the manuscript has greatly improved the logic and elaboration of BFA-triggered GN-association with the exocytosis pathway, in particular as a way to distinguish GN and GNL1 in their localization and functional regulation. The responses to the previous concerns were accepted to be reasonable for this reviewer. Although the story does not necessarily give an exciting mechanistic understanding about how GN functions in development, the establishment of its function domain at the subcellular level (plasma membrane) is an important contribution.Lines 112-113 or 132-133, about the newly generated GN/GNL-GFP lines, more explanations are needed. Since the new localization of GN based on this marker like was the main discovery of the story, it is essential in the main text to explain whether and how this transgene is different or advanced from the previously established GNpro:GN-GFP line that has been widely used in the community and well demonstrated for main association with the GA (Naramoto et al., 2010).

The newly generated lines do not differ from the old marker as much. Both show localization to the same two compartments: the PM and the Golgi apparatus. Importantly, the PM/cortical localization of the old marker appears as punctate foci (Figure 1D, mock and Figure 3A, B) just like those in the new marker, a fact, which we now highlighted in the text.

The finding of Naramoto et al., 2010 that GN localizes to the Golgi is also not really contradicted. This localization is also documented here, except it is proposed that this is not where GN functions in patterning, but only where it functions in secretion together with GNL1, an evolutionarily older, conserved activity of GBF1-type ARF-GEFs (Richter et al., 2007).

The only potential difference between the new and old markers is that the new one decorates the cell periphery in confocal microscopy more frequently. Nonetheless, whether this is a real, relevant difference between the old and new marker remains unclear. For example, clathrin markers are only variably detected at the PM with confocal microscopy, while clathrin-mediated endocytosis is considered a housekeeping process, and TIRF imaging always shows clathrin-coated pits at the PM.

Figure 1B, two panels need label for "weakly fluorescent, small and dynamic structures, probably residing in the cytosol" (Line 140) for both GN-GFP and GNL1-GFP.

We updated Figure 1B to indicate Golgi apparatus and these minor signals, and in the text we referred the reader to Videos 1 and 2, from where these signals can be seen as well.

Lines 150-151, why VAN3 was compared with GN-GFP should be explained in the main text.

The highly dynamic signals of GN-GFP upon BFA treatments are reminiscent of how VAN3 appears in undisturbed conditions. We rewrote this section to better explain the reasoning.

Reviewer #3 (Recommendations for the authors):I am quite satisfied with how you addressed my concerns, and I only have three points left.1. I think you should provide detailed information on how you constructed the GN-GFP line you used throughout this study. For example, you should provide the coordinates of the upstream sequence used, those of the gene or cDNA sequence used, those of the downstream sequence, the precise location of the GFP insertion, the presence of linkers, etc. Alternatively, you could provide the entire sequence of the GN-GFP transgene. This piece of information is essential to this manuscript because so much of what you conclude depends on that construct, and the information you have now provided in Supplementary File 1 is insufficient for researchers to independently replicate your findings.

Thank you for your positive outlook on our work. Perhaps the conclusions do not depend so strongly on the new construct, as we summarized in the response to Reviewer 2. That said, we defined the cloned upstream sequence (GN promoter) in the Methods as -2127 to -1 from the start codon; we indicated that the coding sequence of GN corresponds to that in the reference Araport11 genome; and we provided the information about the linker sequence between GN and GFP, as well as GFP location at the C-terminus of the fusion protein.

2. On ll. 205-220, you wrote: "In our view, even if, following the fwr phenotype, GNfwr is considered as an incompletely effective GN variant, unable of promoting normal LR formation or correct vascular patterning, but functional in embryonic and most of post-embryonic development, the presented argument that PM, rather than the GA, is the site of GN action in developmental patterning, remains valid."I maintain, as I did in my review to the originally submitted manuscript, that the argument is valid only for a subset of the patterning functions of GN, not for all known functions of it. For example, we do not know what the localization is of gn-fwr during vein or lateral-root patterning. Let's suppose we perform that experiment and find that gn-fwr is not localized to the PM during vein or lateral-root patterning. That finding would be consistent with your conclusion that the PM and not the GA is the action of GN's biological function. However, let's suppose that we find that gn-fwr is localized to the PM also during vein or lateral-root patterning. In that case – because gn-fwr has defects in both vein and lateral-root patterning – we would no longer be able to infer that PM-localization is sufficient for all patterning functions of GN. Of course, we would also be unable to infer that PM-localization is not sufficient for those functions of GN since the reduced funcionality of gn-fwr may be due to reasons other than its subcellular localization. Either way, however, the logic of the argument you propose would no longer be sufficiently stringent, and the argument would cease to be necessarily valid. Therefore – I ask – why not be circumspect and propose that it is possible that other functions of GN, untested in this study, may have different localization requirements?

We agree with your reasoning and with leaving open the possibility that functions of GN not tested in this study may have a different localization requirement. We corrected the text to recognize this limitation.

3. Because fwr is a mutant allele of GN, shouldn't it be referred to as gn-fwr and not as GN-fwr throughout the manuscript and figures?

Since the degree of function of this protein is very high, it appeared to us justified to use capital GN, while the *fwr* suffix in lower case still indicates that the protein is mutant. We can think of a somewhat similar example: phosphomutants of various proteins tend to be named with capital letters, as only a limited aspect of their function is modified. Nonetheless, we are happy to follow a consensus of the reviewers, if requested.